# Benchmarking tools for detecting longitudinal differential expression in proteomics data allows establishing a robust reproducibility optimization regression approach

Tommi Välikangas [1], Tomi Suomi [1], Courtney E. Chandler[2], Alison J. Scott [2], Bao Q. Tran[3], Robert K. Ernst [2], David R. Goodlett [4,5] & Laura L. Elo [1,6] ✉

Quantitative proteomics has matured into an established tool and longitudinal proteomics experiments have begun to emerge. However, no effective, simple-to-use differential expression method for longitudinal proteomics data has been released. Typically, such data is noisy, contains missing values, and has only few time points and biological replicates. To address this need, we provide a comprehensive evaluation of several existing differential expression methods for high-throughput longitudinal omics data and introduce a Robust longitudinal Differential Expression (RolDE) approach. The methods are evaluated using over 3000 semi-simulated spike-in proteomics datasets and three large experimental datasets. In the comparisons, RolDE performs overall best; it is most tolerant to missing values, displays good reproducibility and is the top method in ranking the results in a biologically meaningful way. Furthermore, RolDE is suitable for different types of data with typically unknown patterns in longitudinal expression and can be applied by non-experienced users.

In the course of the past few decades, mass spectrometry (MS)-based proteomics has developed significantly and emerged as a powerful tool for clinical biomarker discovery[1]. Currently, MS-powered quantitative proteomics can be considered as an established method routinely used for proteome exploration in biomedical research[2,3].

Longitudinal study designs are generally regarded as having more statistical power to detect differences between the examined study groups than cross-sectional designs[4,5]. While requiring more measurements per individual, less individuals are required to achieve the same statistical power as in cross-sectional studies[4,5]. In addition to having more statistical power, longitudinal study designs deliver information concerning the changes in the studied individuals over

time. In the context of high-throughput transcriptomics, longitudinal experiments for detecting time-resolved gene expression changes have been performed already for more than two decades[6–8]. For proteome profiling, most of the experiments thus far have utilized cross-sectional study designs, but longitudinal proteomics experiments with two[3] or multiple[9,10] time points have begun to emerge.

A recent study comparing longitudinal methods for RNA-sequencing (RNA-seq) gene expression experiments discovered that most of the specific longitudinal methods performed worse than timepoint-wise analysis of the data using traditional pairwise differential expression tools when the number of time points was small (<8)[11]. With only a few time points, most of the tested longitudinal

[1]Turku Bioscience Centre, University of Turku and Åbo Akademi University, FI-20520 Turku, Finland. [2]University of Maryland – Baltimore, Baltimore, MD 21201, USA. [3]US Army 20th Support Command CBRNE Analytical and Remediation Activity, Baltimore, MD 21010-5424, USA. [4]University of Victoria, Victoria, BC V8P 3E6, Canada. [5]International Centre for Cancer Vaccine Science, Gdansk, Poland. [6]Institute of Biomedicine, University of Turku, FI-20520 Turku, Finland. ✉e-mail: laura.elo@utu.fi

methods produced a high number of false positives[11]. When the number of time points was increased, the performance of many of the longitudinal methods also improved[11]. However, as currently relatively few time points are typical in biomedical studies, the usability of such methods which cannot perform well on short time series is limited and new approaches are required.

In the context of proteomics data, another limitation of the longitudinal methods developed for RNA-seq data[11] is that many of them are specifically designed for discrete negative binomially distributed count data and, as such, are not directly applicable. Proteomics data is typically close to normally distributed after logarithm transformation and/or normalization[12], making methods originally proposed for the analysis of longitudinal gene expression microarray data better suited. Among those methods, BETR (Bayesian Estimation of Temporal Regulation)[13] and Timecourse[14] rely on a Bayesian framework, and Microarray Significant Profiles (MaSigPro) builds on a two-step regression strategy[15]. In addition, the popular R package for the analysis of microarray and sequencing data, Limma[16], also contains tools for analyzing longitudinal differential expression. However, since these methods do not take into account the characteristics of MS proteomics data, they might not be optimal for proteomics experiments. In particular, as has been extensively observed in previous studies[17,18], missing values are prevalent in proteomics data and their handling is not trivial[18]. This is particularly the case with the data-dependent acquisition (DDA) label-free proteomics approach, which is popular due to its cost efficiency, speed, and ability to handle complex samples[19]. Furthermore, MS data is prone to noise in quantification[20,21], rendering especially the lowly abundant proteins subject to false positive detections[22].

In addition to the methods specially designed for longitudinal omics data, different statistical modeling approaches have been utilized in their analysis. Several types of linear and non-linear regression-based approaches, with or without random effects, have been applied in various contexts[10,23–25]. For example, Liu et al.[10] used mixed-effects regression modeling with quadratic random terms to detect trends and differential expression patterns in the longitudinal expression of proteins between children developing type 1 diabetes (T1D) and healthy controls. However, no comprehensive comparison on the performance and application of the different approaches exists and, therefore, specific standard practices for analyzing differential expression in longitudinal omics data with regression modeling have not been established. Furthermore, the statistical frameworks developed for the analysis of longitudinal clinical variables might not be best suited for the analysis of longitudinal omics data, in which the number of individuals and the number of measured time points are typically small, but the number of simultaneous variables (e.g., genes or proteins) is very large. Given the complexity of such data and the variety of potential methods available, the selection of an appropriate method is not straightforward.

To facilitate the selection of a suitable method for the discovery of differential expression from longitudinal proteomics data, we present an extensive evaluation of altogether 15 longitudinal approaches and a baseline cross-sectional method, including a method named RolDE (Robust longitudinal Differential Expression) that we develop here. The methods are evaluated in their ability to correctly detect longitudinal differential expression using over 3000 semi-simulated proteomics spike-in datasets with and without missing values, a varying number of time points and including a large variety of linear and non-linear longitudinal trend differences between the examined conditions. In addition, the reproducibility and reliability of the methods, as well as their ability to provide biologically meaningful results are assessed in two large-scale experimental biological datasets. Finally, after demonstrating the robustness and overall good performance of RolDE with various datasets and longitudinal differential expression trends, we apply it to a previously published longitudinal type 1

diabetes proteomics dataset[10] to further illustrate its applicability to non-aligned measurements in a real clinical dataset.

## Results

We extensively evaluated the performance of several approaches to detect differential expression from longitudinal proteomics data: Reproducibility Optimized Test Statistic (BaselineROTS)[26], Bayesian Estimation of Temporal Regulation (BETR)[13], Linear Models for Microarray Data (Limma, LimmaSplines)[16], Timecourse[14], Microarray Significant Profiles (MaSigPro)[15], Extraction of Differential Gene Expression (EDGE)[27], Linear Mixed Model Spline Framework for Analysing Time Course Omics Data (LMMS)[28], Omics Longitudinal Differential Analysis (OmicsLonDA)[29], linear mixed effects regression modeling (Lme), and polynomial mixed effects modeling (Pme). The performance evaluation of the methods was conducted using over 3000 semi-simulated datasets generated on the basis of three spike-in proteomics datasets (UPS1[30], SGSDS[31], and CPTAC[32]) (Fig. 1a). Reproducibility and biological relevance of the findings were further assessed with the large experimental *Francisella tularensis* subspecies *novicida* (*Fn*) dataset (Fig. 1b) and a publicly available dataset on induced human regulatory T (iTreg) cell differentiation[33] (Fig. 1c). In addition to the existing methods, we present RolDE, which is a composite method consisting of three independent modules− RegROTS, DiffROTS and PolyReg−with different approaches for detecting longitudinal differential expression. The combination of these diverse modules allows RolDE to robustly detect varying types of differences in longitudinal trends and expression levels in diverse experimental settings (Fig. 1d). Finally, the ability of RolDE to produce meaningful findings even in data with non-aligned time points was demonstrated using previously published longitudinal T1D proteomics data[10] (Fig. 1e).

### Performance in semi-simulated spike-in proteomics data with single trend categories and no missing values

First, we investigated the performance of the methods in the filtered semi-simulated spike-in datasets, where a single trend category per condition was generated for each dataset (Stable, Linear, LogLike, Poly2, Sigmoid, or PolyHigher; Supplementary Data 1) and only proteins with no missing values were included in the analysis.

In the UPS1-based semi-simulated datasets involving five time points, the overall highest partial areas under the ROC curves (pAUCs) were obtained by RolDE with an interquartile range (IQR) mean pAUC of 0.977 (Fig. 2a), performing better than the second-best method Timecourse with an IQR mean pAUC of 0.973 ($p = 0.059$, one-tailed paired Mann−Whitney $U$ test). The baseline method ROTS that ignores longitudinal trends also performed well with an IQR mean pAUC of 0.941, while the difference to the best performing method RolDE was highly significant ($p < 10^{-15}$). Among the regression-based models, the lower order regression models (denoted by the extension L) performed overall worse than models of higher polynomial degree (denoted by the extension H).

In the SGSDS-based filtered datasets with eight time points and no missing values, all the tested methods performed relatively well (Fig. 2b). Again, the performance of RolDE with IQR mean of pAUC 0.997 was significantly better than the next best methods Limma and LimmaSplines_H ($p < 10^{-11}$).

A closer look at the performance of the methods in the different trend categories suggested that the proposed method RolDE, together with Timecourse and BaselineROTS, performed consistently well in every category in both the UPS1-based and SGSDS-based datasets (Supplementary Fig. 1a, b). The performance of the general regression-based approaches was in concordance with the degree of the regression, as expected. The linear approach Lme performed well when the categories were linear or close to linear; the polynomial regression Pme_L performed better when the examined categories were linear or

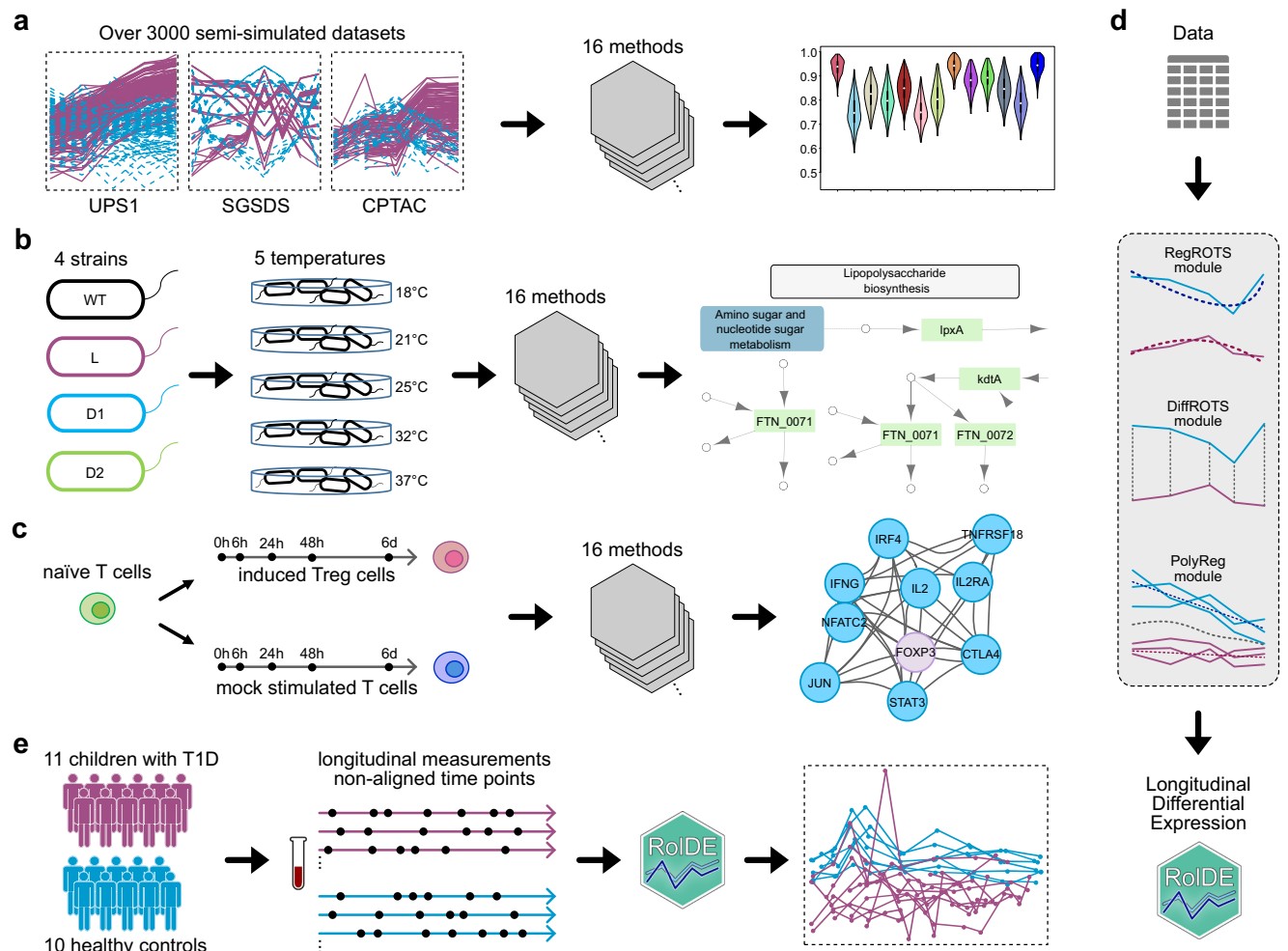

**Fig. 1 | Illustration of the benchmark design and the new RolDE method. a** In total over 3000 semi-simulated spike-in proteomics datasets with two conditions, five (UPS1 and CPTAC) or eight (SGSDS) time points, and varying trend differences between the conditions were generated for an extensive evaluation of the performance of the methods. **b** Reproducibility of the methods across technical replicates and their ability to provide biologically relevant findings in terms of known associated KEGG pathway was evaluated in an experimental *Francisella tularensis* subspecies *novicida (Fn)* dataset. **c** Biological relevance of the findings of the methods was further assessed in a publicly available proteomics data of human induced T regulatory cell (iTreg) differentiation using known Treg-related gene sets from multiple sources. **d** Schematic illustration of the Robust longitudinal Differential Expression (RolDE) method, which is a composite method, consisting of three independent modules with different approaches to detect longitudinal differential expression. The RegROTS module combines individual regression modeling with the power of the established differential expression method Reproducibility Optimized Test Statistic (ROTS). In the DiffROTS module, the expression between all the individuals in the different conditions is directly compared at different time points. The PolyReg module uses polynomial regression modeling to evaluate longitudinal differential expression. The combination of these modules allows RolDE to robustly detect differences in longitudinal trends and expression levels in diverse data types and experimental settings. **e** Ability of RolDE to detect longitudinal differential expression even when the time points in the data are not aligned was further demonstrated using a previously published longitudinal type 1 diabetes proteomics data.

close to second order polynomial. The highest order polynomial regression Pme_H performed well on the broadest spectrum of categories, but the performance was best when the examined categories were of higher order polynomial. The regression spline-based methods EDGE_L and EDGE_H and the OmicsLonDA method were not able to effectively detect longitudinal differential expression when only expression level differences were present between the examined conditions (Stable_Stable category, Supplementary Fig. 1).

**Performance in semi-simulated spike-in proteomics data with single trend categories and including missing values**

Second, we investigated the performance of the methods in the full semi-simulated spike-in datasets with a single trend category per condition but also proteins with missing values included in the analysis. Similarly as with the filtered datasets, which did not involve any missing values, RolDE performed overall best in the full datasets in the presence of missing values (Fig. 2c, d). In the UPS1-based datasets, with

missing values only in the true negative proteins, RolDE performed best with an IQR mean pAUC of 0.976 but with no significant differences in the overall performance to the second-best method Timecourse ($p = 0.257$). In the SGSDS-based full datasets, with missing values also in the true positive spike-in proteins, RolDE and LMMS with IQR mean pAUCs of 0.995 and 0.993, respectively, clearly outperformed the other methods, with RolDE performing significantly better than the second-best method LMMS ($p < 10^{-9}$). BETR and EDGE do not tolerate missing values and were therefore excluded from the analysis of the full datasets.

Investigation of the performance of the methods in the different trend categories suggested that the performance of RolDE, LMMS and BaselineROTS in the full datasets with missing values remained on par to the filtered datasets without missing values (Supplementary Fig. 1c, d), whereas most of the other methods experienced a decrease in their performance across all categories in the SGSDS-based full datasets (Supplementary Fig. 1b, d).

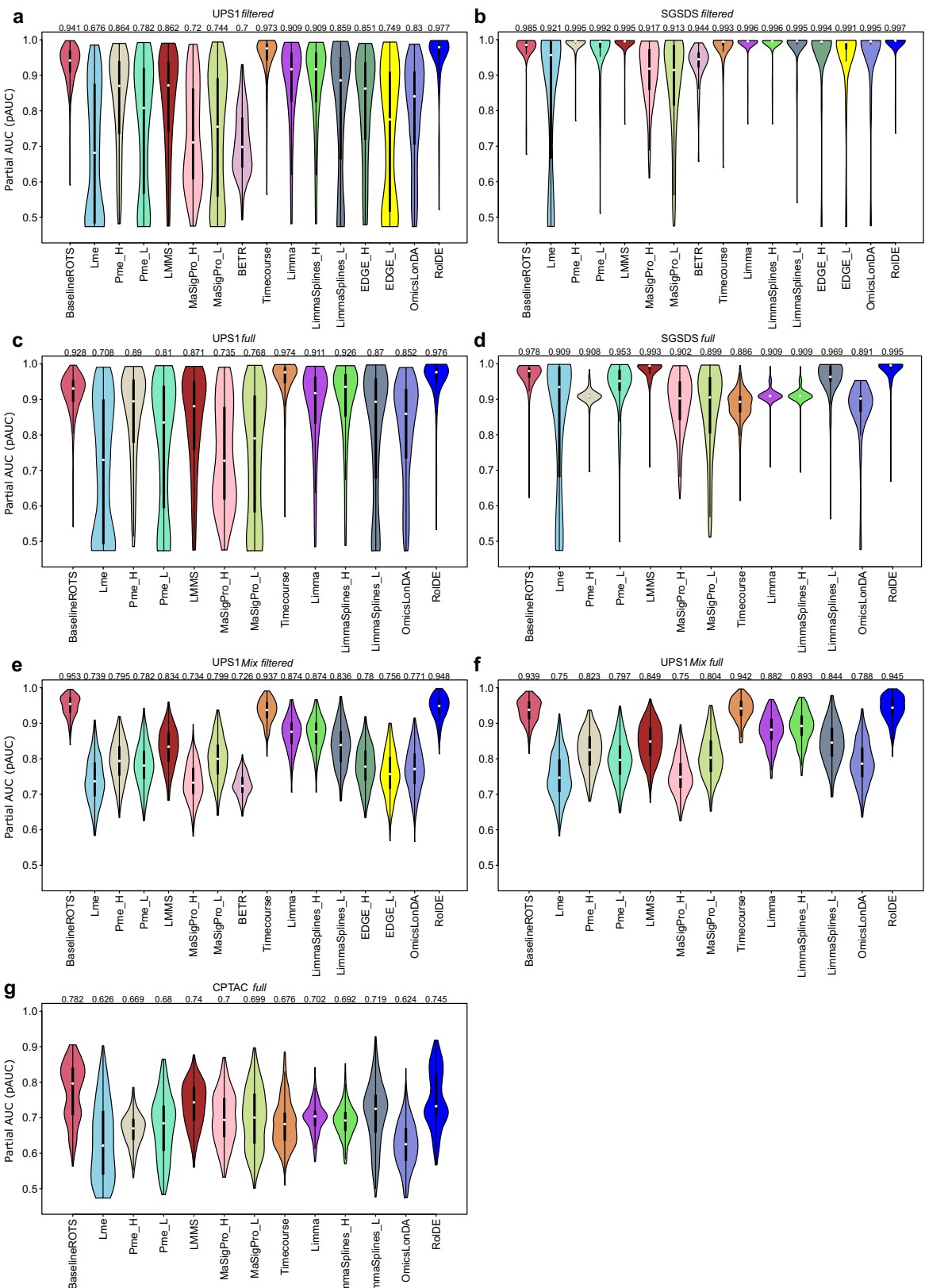

**Fig. 2 | The performance of the examined methods in the semi-simulated spike-in datasets.** **a** UPS1 filtered (*n* = 300 datasets), **b** SGSDS filtered (*n* = 210 datasets), **c** UPS1 full (*n* = 300 datasets), **d** SGSDS full (*n* = 210 datasets), **e** UPS1 Mix filtered (*n* = 300 datasets), **f** UPS1 Mix full (*n* = 300 datasets), **g** CPTAC full (*n* = 300 datasets). The methods were examined in their ability to detect true (known) longitudinal differential expression using receiver operating characteristic (ROC) analysis across datasets with varying longitudinal trend differences in the spike-in proteins (3 replicate samples per condition). The partial areas under the ROC curves (pAUC) between the specificity of 1 and 0.9 were used to measure the performance of the methods. The violin plots display the distribution of pAUCs for each method, including median (white circle), interquartile range (IQR) from the first to third quartile (black box), and 1.5* IQR (whiskers). The IQR mean pAUC for each method is shown above the violin. Each method is shown with a unique color. Source data are provided as a Source Data file.

## Performance in semi-simulated spike-in proteomics data with mixed trend categories

Next, to examine the performance of the methods in more depth, semi-simulated UPS1-based datasets with five time points and mixed trend differences in the spike-proteins were generated to reflect typical real longitudinal proteomics data where proteins with multiple different types of longitudinal trends coexist (UPS1 Mix, Supplementary Data 1).

In the filtered UPS1 Mix datasets without missing values, BaselineROTS with an IQR mean pAUC of 0.953 performed best, followed by RolDE with an IQR mean pAUC of 0.948 ($p < 0.05$, Fig. 2e). Both BaselineROTS and RolDE performed significantly better than the next best method Timecourse with an IQR mean pAUC of 0.937 ($p < 0.05$). Similarly as with the single trend category datasets, the higher order regression models outperformed the lower order models.

In the full UPS1 Mix datasets including missing values, RolDE with an IQR mean pAUC of 0.945 performed best, followed by Timecourse with an IQR mean pAUC of 0.942 ($p = 0.279$, Fig. 2f). Both RolDE and Timecourse significantly outperformed the next best method BaselineROTS with an IQR mean pAUC of 0.939 ($p < 0.05$), while the rest of the methods performed significantly worse, with IQR mean pAUCs below 0.9 in both the filtered and full datasets.

While RolDE displayed the strongest balanced performance over all the trend categories in the UPS1 Mix datasets, also BaselineROTS, Timecourse, Limma, and LimmaSplines_H performed well across the categories (Supplementary Fig. 1e, f). Similarly as with the single-category datasets, the complexity of the trends largely defined in which categories the regression-based approaches performed well; more complex models performed better on a broader spectrum of categories, while the simpler models struggled when the polynomial complexity of the trends increased. Overall, proteins with Linear_Sigmoid and Linear_LogLike trend differences were most challenging across all the methods.

## Effect of missing values in the datasets

The original UPS1 and SGSDS datasets contained moderate to low numbers of missing values (14.1% and 3.4%, respectively) and only few missing values in the spike-in proteins (0% and 4.2%, respectively). Therefore, to further push the methods, semi-simulated datasets with a larger proportion of missing values were generated using the CPTAC dataset, which had a larger overall proportion of missing values (19.5%) and a considerably larger proportion of missing values in the spike-in proteins (29.4%) than the UPS1 and SGSDS datasets. In the presence of such a large proportion of missing values, all the methods performed clearly worse than in the UPS1 and the SGSDS datasets (Fig. 2g). The cross-sectional method BaselineROTS applied separately to each time point significantly outperformed the longitudinal methods with an IQR mean pAUC of 0.782, followed by RolDE with an IQR mean pAUC of 0.745 ($p < 10^{-5}$). Both BaselineROTS and RolDE performed significantly better than the next best method LMMS with an IQR mean pAUC of 0.740 ($p < 10^{-5}$).

Furthermore, many of the evaluated methods struggled to provide valid scores for proteins in the presence of missing values in the data. To evaluate how consistently the different methods were able to provide scores for the proteins in the datasets, IQR mean proportions of valid scores for each method and dataset type were calculated (Table 1). In addition to RolDE, which by default generates a ranking for all the proteins in a dataset, the linear regression method Lme and the linear mixed model spline method LMMS were most often able to provide a valid ranking for the proteins in all the datasets. Overall, the lower order models, Pme_L and LimmaSplines_L, were able to provide a ranking more often than their higher order counterparts Pme_H and LimmaSplines_H.

## Effect of reduced number of time points

Since it has remained common to have only few time points in real biomedical studies, we also evaluated the performance of the best-performing methods RolDE, Timecourse, Limma, and the baseline ROTS in additional 1200 semi-simulated datasets with only three or

**Table 1 | Median proportions of missing values and interquartile range (IQR) mean proportions of missing valid result scores in the semi-simulated spike-in datasets and the experimental *Francisella tularensis* subspecies *novicida* dataset**

| | UPS1 Filtered | SGSDS Filtered | UPS1Mix Filtered | UPS1 Full | SGSDS Full | CPTAC Full | UPS1Mix Full | WT vs. L | WT vs. D2 | WT vs. D1 |
|---|---|---|---|---|---|---|---|---|---|---|
| **Missing values in proteins** | | | | | | | | | | |
| All proteins | 0.0% | 0.0% | 0.0% | 14.1% | 3.3% | 19.6% | 14.0% | 7.0% | 7.7% | 7.1% |
| Spike-in proteins | 0.0% | 0.0% | 0.0% | 0.0% | 4.6% | 28.5% | 0.0% | | | |
| **Missing values in results** | | | | | | | | | | |
| BaselineROTS | 0.0% | 0.0% | 0.0% | 11.0% | 1.8% | 15.2% | 9.0% | 2.0% | 1.8% | 2.3% |
| Lme | 0.0% | 0.0% | 0.0% | 7.7% | 0.5% | 7.4% | 7.2% | 1.5% | 1.4% | 1.3% |
| Pme_H | 0.0% | 0.0% | 0.0% | 19.3% | 3.2% | 19.5% | 19.1% | 13.2% | 16.1% | 12.7% |
| Pme_L | 0.0% | 0.0% | 0.0% | 11.9% | 1.2% | 11.4% | 11.2% | 2.8% | 2.7% | 2.7% |
| LMMS | 0.0% | 0.0% | 0.0% | 4.8% | 0.6% | 5.9% | 4.0% | 0.8% | 0.7% | 0.8% |
| MaSigPro_H | 20.3% | 22.8% | 23.2% | 30.6% | 24.1% | 40.6% | 32.9% | 29.3% | 31.4% | 16.7% |
| MaSigPro_L | 33.6% | 35.7% | 46.0% | 41.1% | 36.5% | 52.2% | 51.8% | 43.6% | 51.2% | 26.8% |
| BETR | 0.0% | 0.0% | 0.0% | NA | NA | NA | NA | 29.6% | 34.1% | 32.2% |
| Timecourse | 0.0% | 0.0% | 0.0% | 23.4% | 6.9% | 27.8% | 24.4% | 20.8% | 25.0% | 23.5% |
| Limma | 0.0% | 0.0% | 0.0% | 19.3% | 3.2% | 19.2% | 19.0% | 13.2% | 16.1% | 12.7% |
| LimmaSplines_H | 0.0% | 0.0% | 0.0% | 19.3% | 3.2% | 19.2% | 19.0% | 13.2% | 16.1% | 12.7% |
| LimmaSplines_L | 0.0% | 0.0% | 0.0% | 11.9% | 1.2% | 11.1% | 11.1% | 2.7% | 2.7% | 2.6% |
| EDGE_H | 0.0% | 0.0% | 0.0% | 34.2% | 15.0% | 50.7% | 33.8% | 29.6% | 34.1% | 32.2% |
| EDGE_L | 0.0% | 0.0% | 0.0% | 34.3% | 15.0% | 50.7% | 33.8% | 29.6% | 34.1% | 32.2% |
| OmicsLonDA | 0.0% | 0.0% | 0.0% | 21.8% | 4.3% | 25.6% | 21.6% | 11.6% | 16.2% | 13.4% |
| RolDE | 0.0% | 0.0% | 0.0% | 0.0% | 0.0% | 0.0% | 0.0% | 0.0% | 0.0% | 0.0% |

NA refers to no scores delivered at all by the method. Source data are provided as a Source Data file.

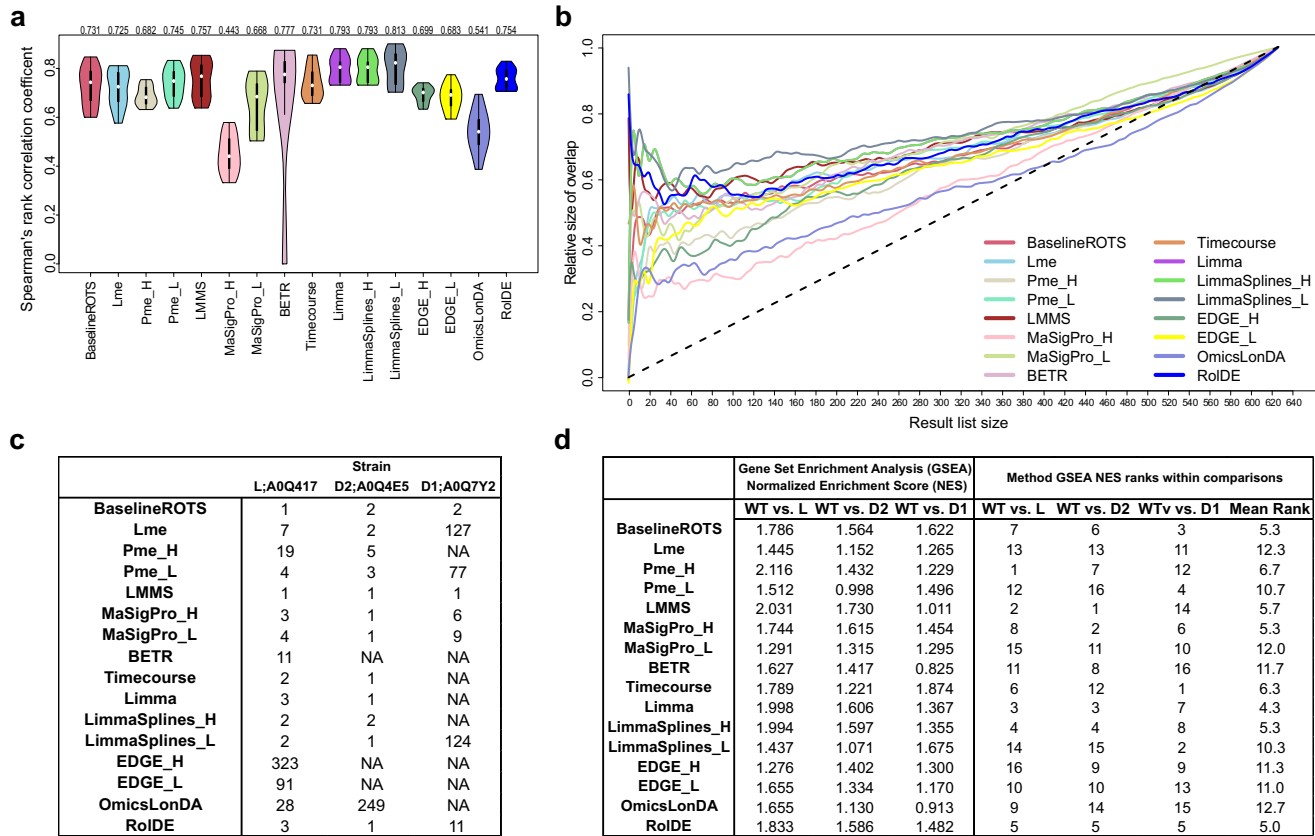

**c**

| | Strain | | |
|---|---|---|---|
| | L;A0Q417 | D2;A0Q4E5 | D1;A0Q7Y2 |
| **BaselineROTS** | 1 | 2 | 2 |
| **Lme** | 7 | 2 | 127 |
| **Pme_H** | 19 | 5 | NA |
| **Pme_L** | 4 | 3 | 77 |
| **LMMS** | 1 | 1 | 1 |
| **MaSigPro_H** | 3 | 1 | 6 |
| **MaSigPro_L** | 4 | 1 | 9 |
| **BETR** | 11 | NA | NA |
| **Timecourse** | 2 | 1 | NA |
| **Limma** | 3 | 1 | NA |
| **LimmaSplines_H** | 2 | 2 | NA |
| **LimmaSplines_L** | 2 | 1 | 124 |
| **EDGE_H** | 323 | NA | NA |
| **EDGE_L** | 91 | NA | NA |
| **OmicsLonDA** | 28 | 249 | NA |
| **RolDE** | 3 | 1 | 11 |

**d**

| | Gene Set Enrichment Analysis (GSEA) Normalized Enrichment Score (NES) | | | Method GSEA NES ranks within comparisons | | | |
|---|---|---|---|---|---|---|---|
| | WT vs. L | WT vs. D2 | WT vs. D1 | WT vs. L | WT vs. D2 | WTv vs. D1 | Mean Rank |
| **BaselineROTS** | 1.786 | 1.564 | 1.622 | 7 | 6 | 3 | 5.3 |
| **Lme** | 1.445 | 1.152 | 1.265 | 13 | 13 | 11 | 12.3 |
| **Pme_H** | 2.116 | 1.432 | 1.229 | 1 | 7 | 12 | 6.7 |
| **Pme_L** | 1.512 | 0.998 | 1.496 | 12 | 16 | 4 | 10.7 |
| **LMMS** | 2.031 | 1.730 | 1.011 | 2 | 1 | 14 | 5.7 |
| **MaSigPro_H** | 1.744 | 1.615 | 1.454 | 8 | 2 | 6 | 5.3 |
| **MaSigPro_L** | 1.291 | 1.315 | 1.295 | 15 | 11 | 10 | 12.0 |
| **BETR** | 1.627 | 1.417 | 0.825 | 11 | 8 | 16 | 11.7 |
| **Timecourse** | 1.789 | 1.221 | 1.874 | 6 | 12 | 1 | 6.3 |
| **Limma** | 1.998 | 1.606 | 1.367 | 3 | 3 | 7 | 4.3 |
| **LimmaSplines_H** | 1.994 | 1.597 | 1.355 | 4 | 4 | 8 | 5.3 |
| **LimmaSplines_L** | 1.437 | 1.071 | 1.675 | 14 | 15 | 2 | 10.3 |
| **EDGE_H** | 1.276 | 1.402 | 1.300 | 16 | 9 | 9 | 11.3 |
| **EDGE_L** | 1.655 | 1.334 | 1.170 | 10 | 10 | 13 | 11.0 |
| **OmicsLonDA** | 1.655 | 1.130 | 0.913 | 9 | 14 | 15 | 12.7 |
| **RolDE** | 1.833 | 1.586 | 1.482 | 5 | 5 | 5 | 5.0 |

**Fig. 3 | Reproducibility and biological relevance of the examined longitudinal differential expression methods in the experimental *Francisella tularensis* subspecies *novicida (Fn)* proteomics data. a** Reproducibility of the methods was evaluated using the Spearman's rank correlation coefficient between technical replicate result lists ($n = 18$ correlations for each method over all the six possible pairwise comparisons of strains with three technical repeats for each comparison). The violin plots display the distribution of correlations for each method, including median (white circle), interquartile range (IQR) from the first to third quartile (black box), and 1.5* IQR (whiskers). The IQR mean of the correlations for each method is shown above each violin. Each method is shown with a unique color. **b** Median proportional overlaps of the top $k$ findings between the technical replicate result lists when $k$ was varied from 1 to the length of the entire dataset. Proteins with missing values were filtered out from the dataset prior to the reproducibility testing in **a** and **b**. **c** Rankings of the proteins related to the modified acyltransferases when longitudinal differential expression was examined by each method in each pairwise comparison between the null mutant strains LpxD1 (D1), LpxD2 (D2), LpxL (L) and the wild type (WT) over the five temperatures. The UniProt accession of each protein related to the modified null mutant is shown. NA refers to not detected at all by the method. **d** The normalized enrichment scores (NES) from the gene set enrichment analysis (GSEA) of the relevant Lipopolysaccharide synthesis pathway and the associated knockout pathway proteins among the findings of the different methods in comparisons of the acyltransferase null mutant strains and the wild type. The NES scores of the methods were ranked within each comparison and a mean rank was calculated over all the comparisons for each method. Source data are provided as a Source Data file.

four time points. Altogether 300 filtered (without missing values) and 300 full (including missing values) UPS1-based semi-simulated datasets of both lengths were explored. Again, RolDE performed overall best, significantly outperforming the second-best method Timecourse ($p < 0.02$ in all scenarios, Supplementary Fig. 2). Both RolDE and Timecourse performed better than the cross-sectional baseline method ($p < 0.001$ in all scenarios) and Limma ($p < 10^{-15}$). None of the methods were largely affected by the missing values in the true negative proteins.

### Reproducibility between technical replicate *Fn* datasets

To evaluate the longitudinal differential expression methods in a real proteomics study setting, we generated an experimental membrane-enriched longitudinal proteomics data of *Francisella tularensis* subspecies *novicida (Fn)* (Fig. 1b), including four strains: the wild type (WT) and null mutants of lpxD1 (D1), lpxD2 (D2) and lpxL (L). For evaluation of the methods, differential expression between the null mutant and the wild-type strains was investigated, while their growth at five different temperatures offered a surrogate for the different time points. Three technical replicates of the three biological replicate samples were used to form technical replicate datasets analyzed with the

different methods to evaluate the reproducibility of the methods. To examine the overall reproducibility of the results produced by the tested methods, all possible pairwise combinations of the strains were analyzed (WT vs. D1, WT vs. D2, WT vs. L, D1 vs. D2, D1 vs. L, D2 vs. L).

The overall proportion of missing values in the *Fn* data was 10.8%, being highest in the 37 °C samples with more than 25% of all the values missing in all strains but L (Supplementary Fig. 3). The proportion of valid rankings provided by the different methods were consistent with the full semi-simulated spike-in benchmark datasets with missing values (Table 1): RolDE, Lme, BaselineROTS, Pme_L, LimmaSplines_L, and LMMS provided a ranking for more than 97% of all the proteins in each pairwise comparison; Limma, LimmaSplines_H, and Pme_H provided a ranking for 84–87% of the proteins; OmicsLonDA was able to rank 84–88% of the proteins; and Timecourse, EDGE and MaSigPro provided a ranking for <80% of the proteins.

Spearman correlations between the technical replicate lists were on average highest with different variants of Limma (IQR mean correlations 0.793–0.813, Fig. 3a), followed by LMMS (IQR mean correlation 0.757) and RolDE (IQR mean correlation 0.754), both of which had significantly higher correlations than the next method Pme_L with an IQR mean correlation of 0.745 ($p < 0.05$). In addition to the overall

correlations of the result lists, we examined the median overlaps of the top findings between the replicate datasets, when the size of the overlap was varied from 1 to the total number of proteins in the dataset. In line with the correlation results, the different variants of Limma, LMMS, and RolDE showed a high overlap between the technical replicate result lists, especially among the top findings (Fig. 3b).

## Biological relevance of the findings in the *Fn* data

Typically, the top findings in any experiment are the most interesting ones and will most likely be selected for further validation. Therefore, consistency of a method in delivering the same findings at the top of the list is a highly desirable quality. However, reproducibility alone does not guarantee biological relevance. Therefore, we also examined how the different methods detected the proteins related to the modified acyltransferases in the *Fn* data. For this purpose, longitudinal differential expression over the temperatures was examined pairwise between the wild type (WT) and the null mutants of lpxD1 (D1), lpxD2 (D2) and lpxL (L) in the full data with technical replicates averaged for each biological replicate. The result lists from each method were ordered based on the strength of differential expression given by the method.

When investigating the acyltransferases themselves, most of the methods ranked the proteins related to the acyltransferases lpxL and lpxD2 to the top of their result lists, with the exception of BETR, EDGE_H, and EDGE_L, which failed to provide a score for lpxD2. More variation was observed in how the methods ranked the protein related to lpxD1, with BETR, Pme_H, Timecourse, Limma, LimmaSplines_H, EDGE_H, EDGE_L and OmicsLonDA failing to provide a score for the protein (Fig. 3c).

In addition to the acyltransferases themselves, we also investigated how the different methods ranked the related pathway of the synthesis of lipid A and the endotoxin lipopolysaccharide (LPS). Accordingly, we examined how the relevant proteins in different comparisons in the KEGG pathway Lipopolysacchararide biosynthesis −*Francisella tularensis* subsp. *novicida* U112 (ftn00540) and the associated Lipopolysaccharide biosynthesis knockout pathway (ko00540) were detected in our *Fn* dataset (Fig. 3d, Supplementary Table 1). In each comparison of the wild type and mutant strains, RolDE, Limma, LimmaSplines_H, and BaselineROTS consistently provided top findings with high biological relevance (Fig. 3d). Also MaSigPro_H, LMMS, and Timecourse provided highly biologically relevant top results in most comparisons.

## Biological relevance of the findings in the human iTreg data

In addition to evaluating the biological relevance of the findings of the different methods in the *Fn* data, an additional publicly available dataset of human iTreg in vitro induction[33] was included in the comparison to comprehensively assess the methods in delivering biologically meaningful results. In this dataset of almost 10,000 proteins, naïve T cells were stimulated and iTreg cells were induced using two different protocols. Proteome profiles were acquired at multiple time points during iTreg induction or mock stimulation at 6 h, 24 h, 48 h, and 6 d. Longitudinal differential expression was determined between the iTreg and the mock-stimulated control cells, separately for both protocols, resulting in two comparisons.

As the exact proteins afflicted by the induction process are still under research[33,34] and not known in detail, several Treg related gene sets were used to assess the biological relevance of the findings. Altogether 28 Treg cell-related human gene sets were downloaded from the Molecular Signatures Database (MSigDB version 7.5.1)[35] and combined into 14 Treg gene sets (Supplementary Data 2). In addition, a Treg signature gene set[34] from a large human study, genes from the relevant Reactome pathway (RUNX1 and FOXP3 control the development of regulatory T lymphocytes), and the interactome of the forkhead box P3 (FOXP3) protein, which is a lineage specification factor of Treg cells, were used to explore Treg-related functional enrichment. In both of the comparisons using the two different iTreg induction protocols, the highest enrichments were most consistently detected with RolDE, BaselineROTS, and BETR (Fig. 4a), suggesting that these methods were able to provide biologically relevant results and findings related to the Treg cell state expected using the Treg cell induction protocols.

## Overlaps of the findings in the human iTreg data

To explore the reliability of the findings with the different methods in the experimental iTreg data, we compared the overlaps of the top findings between the methods. Common findings by multiple methods are often considered more reliable than those detected only by a single method[11,33]. When comparing the top 1000 findings of the overall best-performing methods in this study (RolDE, Timecourse, Limma, LMMS, and BaselineROTS), RolDE had the largest number of shared and the lowest number of unique findings in both comparisons (Fig. 4b); RolDE had only 94 and 90 unique findings in the two comparisons, compared to Limma with 112 and 133, BaselineROTS with 177 and 559, LMMS with 278 and 262, and Timecourse with 288 and 321 unique findings.

To further expand these investigations, we explored the proportion of the top 1000 findings of each method that was common with the other methods in their respective top 1000 findings over the two comparisons (Fig. 4c). Here, only the best-performing variant of each method was included for an unbiased and accurate comparison, as for methods with multiple variants, a large proportion of the top result list is shared between the variants. Overall, BETR had the lowest proportion of unique detections, followed closely by EDGE_H, RolDE, LMMS, and Pme_H. Less than 10% of their top 1000 findings were unique to the method and over 50% of the findings were shared with six or more other methods.

## Application of RolDE to longitudinal type 1 diabetes proteomics data with non-aligned time points

While the semi-simulated spike-in datasets and the experimental *Fn* and iTreg datasets contained perfectly aligned time points (or their surrogates), this is not always the case in various real experimental settings. Therefore, we demonstrate the applicability of the overall best-performing RolDE method in a previously published longitudinal type 1 diabetes proteomics data with non-aligned time points[10]. The dataset contains blood plasma protein expression measurements of 11 children developing T1D and 10 matched controls[10]. Nine longitudinal samples were collected for each child with non-aligned time points. The aim was to detect early T1D related biomarkers from the blood even before seroconversion and T1D diagnosis, which would allow earlier disease prediction and intervention.

RolDE detected a total of 15 proteins with longitudinal differential expression between the T1D cases and controls at false discovery rate (FDR) of 0.05 (Supplementary Table 2). These included proteins with clear trend differences as well as proteins with clear expression level differences between the cases and controls. Two keratins, K1H1 and KRT86, were detected as highly differentially expressed (Fig. 5a, Supplementary Table 2). Decreased levels of keratins[36] as well as increased keratin metabolism[37] have been previously associated with diabetes and hyperglycemia with keratinocyte proliferation, differentiation, and function[38]. Downregulation of another detected protein (Fig. 5b), cell growth regulator with EF-hand domain 1 (CGRE1), has been associated with hyperglycemia in humans[39]. Significant decrease in serotransferrin (TRFE) in T1D cases was also detected (Fig. 5c). Serotransferrin has been associated with diabetes and glucose metabolism levels in multiple studies[40–42]; Metz et al. reported serotransferrin to be downregulated by two-fold in the plasma of T1D patients compared to controls[42]. Similarly, serum Amyloid A1 (SAA1, Fig. 5d), a top differential expression finding of RolDE, has been previously linked to diabetes and blood glucose levels in multiple studies[43–45]. Furthermore, sodium channel and clathrin linker 1 (SCLT1)

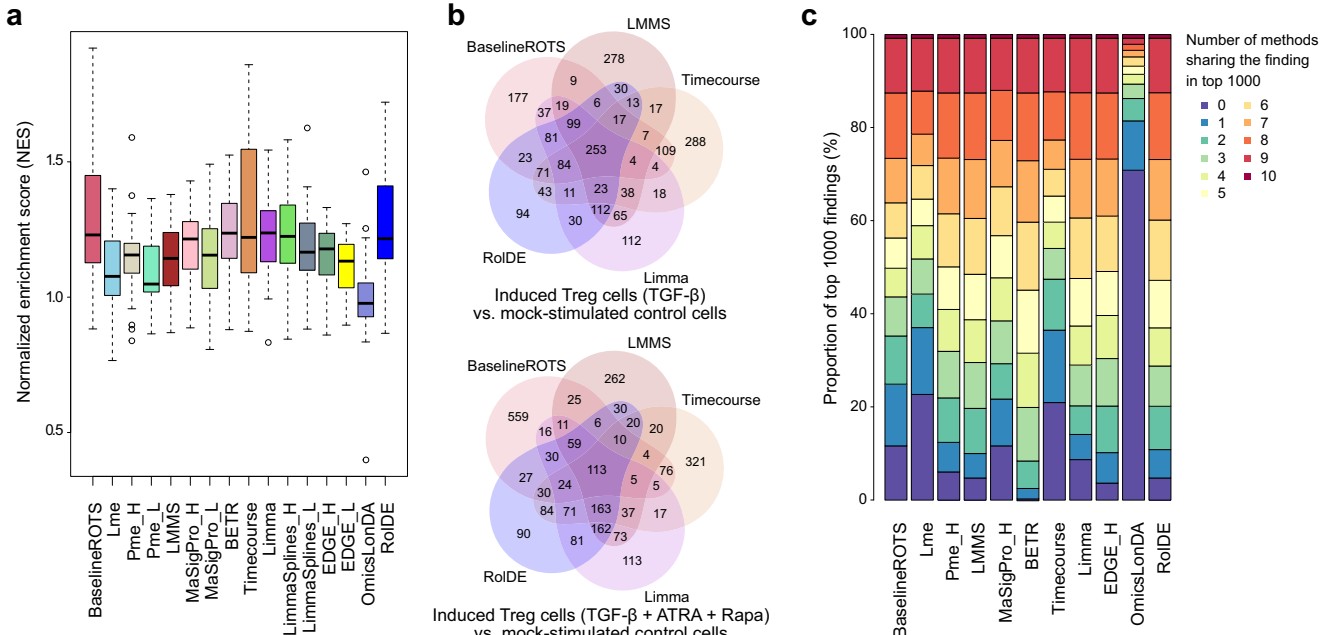

**Fig. 4 | Biological relevance and shared findings between the methods in the experimental human-induced T regulatory (iTreg) cell data. a** The normalized enrichment scores (NES) from the gene set enrichment analysis (GSEA) of 17 known Treg-related gene sets over comparisons of the two different induction protocols and the mock-stimulated control cells (*n* = 3 biologically independent samples in each comparison group, 34 NES values for each method). The boxplots display the NES scores for each method, including median (horizontal black line), interquartile range (IQR) from the first to third quartile (black box), and 1.5* IQR (whiskers). Each method is shown with a unique color. **b** Overlaps of the top 1000 findings with the five best performing methods in the two comparisons using the different induction protocols; with and without all-trans retinoic acid (ATRA) and rapamycin (Rapa). **c** Proportions of the top 1000 findings for each method detected also by different numbers (colors) of other methods in their top 1000 over the comparisons with both protocols. Source data are provided as a Source Data file.

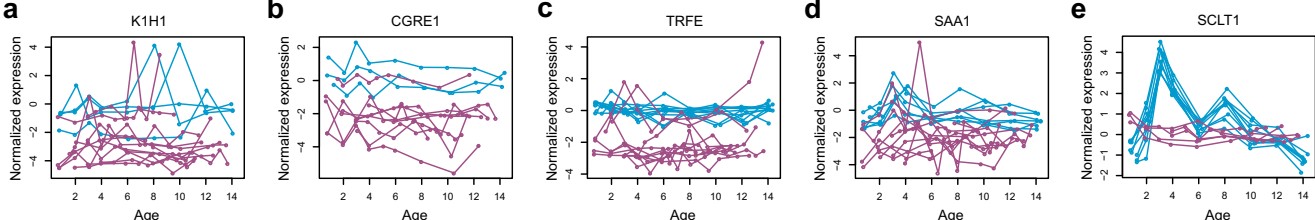

**Fig. 5 | Longitudinal expression of representative examples of the top differential expression detections by the Reproducibility optimized longitudinal Differential Expression method RolDE in the longitudinal type 1 diabetes blood plasma proteomics data of Liu et al.[10]. a** Type I cuticular keratin (K1H1), **b** Cell growth regulator with EF hand domain protein 1 (CGRE1), **c** Serotransferrin (TRFE), **d** Serum amyloid A-1 protein (SAA1), **e** Sodium channel and clathrin linker 1 (SCLT1). Blue color represents individuals in the control group (*n* = 10) while red color corresponds to children in the type 1 diabetes group (*n* = 11). Source data are provided as a Source Data file.

was detected as highly significant (Fig. 5e). It has been linked to ciliopathies in human[46] and many ciliopathies have been linked to obesity and insulin secretion in the pancreas by the regulating effect of pancreatic cilia[47].

Finally, we compared the RolDE results to the results from the original study by Liu et al.[10]. The study detected two proteins with differential expression between T1D cases and controls at FDR of 0.05 using mixed effects modeling with linear and quadratic random effects in addition to the fixed effects[10]. One of these, SCLT1, was also detected by RolDE, whereas the other one, carbonyl reductase 1 (CBR1), had FDR of ~0.07 with RolDE (Supplementary Fig. 4).

Taken together, these top findings demonstrate the ability of RolDE to detect proteins with various kinds of differential expression patterns also in data with non-aligned time points.

## Discussion

We have comprehensively evaluated the performance of 16 approaches, including the RolDE method developed in this study, for

detecting differential expression between two conditions in longitudinal proteomics data. Altogether 3120 semi-simulated proteomics datasets were used to benchmark the ability of the different approaches to detect the truly differentially expressed spike-in proteins with various longitudinal expression patterns and differences, both with and without missing values present in the data. The reproducibility and biological relevance of the results of the different methods were further evaluated using a longitudinal experimental dataset based on mutants of *Francisella tularensis* subspecies *novicida* at various temperatures known to change the expression of proteins in the Raetz pathway responsible for Lipid A Biosynthesis[48]. In addition, the biological relevance of the findings was assessed in a publicly available time-series dataset[33] of human iTreg cell differentiation. Finally, we demonstrated the ability of RolDE to detect differential expression even in data where the time points in different conditions are not aligned using a previously published T1D dataset[10].

Overall, the proposed longitudinal differential expression method RolDE performed best in the semi-simulated spike-in datasets,

including datasets with missing values in the true positive spike-in proteins (Fig. 2d, g). Missing values are a typical occurrence in proteomics data, especially in the popular label-free data-dependent acquisition approach. Furthermore, as the proteins of interest are typically unknown beforehand and may also contain missing values (such as in the SGSDS datasets or in the *Fn* dataset), the ability of a differential expression method to deal with versatile missing values might be crucial in detecting relevant proteins to the experimental question. RolDE also displayed the most balanced performance of detecting all possible types of trend differences with good consistency (Supplementary Fig. 1). It is typical that a researcher does not know beforehand what types of differences in longitudinal expression are to be expected. The ability of a method to detect various types of trend differences is thus essential for a comprehensive analysis of the data.

The overall reproducibility of the results in the biological *Fn* data as well as the reproducibility of the top results was best with the Limma-based approaches, RolDE and LMMS. The relevance of the top findings in terms of both the modified proteins themselves as well as the directly associated KEGG Lipopolysaccharide Biosynthesis and Lipopolysaccharide Biosynthesis knockout pathway proteins was most consistent with RolDE, Limma, LimmaSplines_H, and BaselineROTS. LPS is an essential part of the Gram-negative bacteria outer membrane and forms an amphipathic interface between the bacteria and the environment[49,50]. LPS is composed of three structural parts: the core polysaccharide, O-antigen, and the membrane anchor lipid A[49]. Modifications of lipid A can alter the structure and pathogenicity of the bacterium and are influenced by environmental conditions such as temperature[51]. In *Fn*, temperature has been shown to alter the composition of lipid A[51]. The chosen KEGG pathway focusing on LPS biosynthesis therefore served as valuable readout for exploring and validating the performance of the different methods and also highlight the consistency of the generated proteomics data within our understanding of *Fn* biology field as a whole.

In the experimental human iTreg data, the overall biological relevance of the results was highest with RolDE, BaselineROTS, BETR, and Timecourse, as assessed by their enrichment with multiple related gene sets from various sources, which were expected to provide a good surrogate of what should be affected during the induction process. The results were well in line with the semi-simulated spike-in datasets as well as the *Fn* data, where especially RolDE and BaselineROTS performed overall best. The relevance of the RolDE findings was further supported by the observation that it was among the methods with the largest proportion of shared and the lowest proportion of unique findings among its top 1000 detections. For instance, in a previous comparative study of differential expression methods in RNA-seq, it has been observed that true positives are generally shared by the methods, while the unique findings by a single method are rarely true positive detections[11].

In summary, the proposed method RolDE performed overall best. In addition to excellent performance in the semi-simulated spike-in datasets, it had good overall and especially top reproducibility (Fig. 3a, b), was among the top methods in the experimental *Fn* (Fig. 3c, d) and iTreg datasets (Fig. 4a), as well as displayed a high degree of overlap of its top findings in the iTreg data (Fig. 4b, c). Furthermore, RolDE delivered interesting findings even when the time points in the data were not aligned (Fig. 5). Even though not always being the best method in each explored scenario, the consistently good performance throughout the various datasets and comparison aspects suggests RolDE to offer robust general all-round approach for detecting longitudinal differential expression. By default, RolDE provides a score for all the proteins in the data.

The ability of RolDE to deal with missing values and noisy measurements can, for the most part, be credited to four features of the method: (1) modularity, (2) application of ranks and rank products, (3) application of longitudinal regression modeling, and (4)

reproducibility-optimization with ROTS. The benefit of modularity is that, even if some modules would not be able to provide a valid score for a protein, the other modules with diverse approaches can still provide enough evidence for detecting possible differential expression, whereas proteins with multiple poor ranks will end up towards the end of the result list. The motivation to use ranks and rank products in RolDE is to avoid relying on any distributional assumptions for the data. The rank product approach has been previously shown to be especially effective for noisy datasets and for small numbers of replicates[52,53]. An additional benefit of the longitudinal regression modeling applied in the RegROTS and PolyReg modules is their tolerance against missing values. Although the method does not use regression for any imputation of missing values as such, the longitudinal regression models can typically be estimated for a protein even in the presence of some missing values. The coefficients of these models are then used to explore longitudinal differential expression. Similarly, most of the other longitudinal methods tested in this study also used some form of regression modeling. Finally, the benefit of the reproducibility-optimization procedure of ROTS is its ability to produce robust results, as shown by us and others in this study and various other omics studies[12,30,54–56]. In addition, ROTS tolerates a moderate number of missing values in the data matrix by effectively ignoring their contribution during the operation of the procedure[26], making it particularly suitable for proteomics data.

While RolDE displayed good performance in all scenarios, most of the methods performed well in at least some comparison. Similar to RolDE, Timecourse, a longitudinal differential expression method developed for microarrays and utilizing a Bayesian framework, performed excellent in the semi-simulated datasets and produced biologically relevant results in the iTreg data and mostly in the *Fn* data. However, missing values in the spike-in proteins clearly hindered the performance of the method and Timecourse was able to provide a ranking only for ~72%–79% of proteins in the UPS1 and CPTAC semi-simulated datasets and the experimental *Fn* data when missing values were present (Table 1). Furthermore, Timecourse was not able to deliver a ranking for all the modified acyltransferases known to change between the conditions in the *Fn* dataset. A method that is not able to deal with different kinds of missingness in the data might be impractical in the context of proteomics data, where missing values can occur due to various reasons[18,57]. In addition, Timecourse does not provide significance estimates for the features, which might also limit its usability.

Although not quite on par with RolDE, the different variants of Limma and LMMS offered consistent overall performance as well. In general, LMMS performed well in the semi-simulated spike-in datasets, especially in SGSD-based datasets with more time points. Similar to RolDE, LMMS was tolerant against missing values and typically provided scores for most of the proteins in the datasets (Table 1). In addition, LMMS had good reproducibility and biological relevance of the results, especially in the *Fn* data. Limma provides two different approaches to detect longitudinal differential expression. In the semi-simulated spike-in datasets, the group mean parametrization approach (Limma) performed similar to the cubic splines approach (LimmaSplines) when the data did not contain missing values. Some differences in the performance could be observed when missing values were present in the data. In particular, the LimmaSplines_L approach with less degrees of freedom was best among the Limma-based approaches when missing values were present also in the spike-in proteins in the SGSDS and CPTAC semi-simulated datasets (Fig. 2), but not in datasets without missing values or when missing values were present only in the background proteins (UPS1-based datasets). Reproducibility of the results was very high in all variants of Limma and in particular the group mean parametrization approach (Limma) and the splines variant with more degrees of freedom (LimmaSplines_H) provided biologically relevant results in the *Fn* data. However, similar

to Timecourse, Limma and LimmaSplines_H failed to provide a score for all the modified acyltransferases in the *Fn* data.

The cross-sectional differential expression method used in this study as a baseline, the BaselineROTS, also performed well in the comparisons. BaselineROTS displayed good performance in the semi-simulated spike-in datasets, especially in the UPS1 Mix datasets, where several different types of longitudinal trends and trend differences were present. Furthermore, BaselineROTS displayed good overall reproducibility and was able to provide biologically meaningful results in both of the examined experimental datasets. Overall, when considering all the examined scenarios, out of the longitudinal approaches, only RolDE consistently outperformed BaselineROTS. Also Timecourse and the Limma-based approaches outperformed or performed on par with BaselineROTS in most explored scenarios. These results suggest that a well-performing two group differential expression method applied separately on each time point and with the results summarized with a suitable metric (e.g., with minimum of the significance values as in this study) can sometimes provide a meaningful ranking for the strength of longitudinal differential expression in the data. Our results are in line with the previous results of a study comparing longitudinal methods for RNA-sequencing, where most longitudinal methods performed worse than pairwise differential expression approaches applied timepoint-wise[11].

BETR and variants of EDGE did not tolerate missing values and did not match the performance of RolDE, Timecourse and Limma in the semi-simulated datasets and in terms of reproducibility, but fared better on the experimental iTreg data, especially BETR. While performing decently in many of the tested semi-simulated datasets, OmicsLonDa struggled in many scenarios and in the iTreg data most of its top 1000 findings were unique. However, OmicsLonDa was initially developed for identification of time intervals with significant differences between the conditions in longitudinal data and is perhaps not suitable as a general tool to detect any longitudinal differential expression between the conditions. Similarly, by default MaSigPro explores the overall time-associated changes over both conditions, but these effects were not included in this study, as the interest was explicitly in the longitudinal differential expression between the conditions.

There was considerable variation in the performance of the various regression modeling approaches tested. Overall, the higher order polynomial regression models outperformed the lower degree models. Only in the SGSDS full datasets, the overall performance of Pme_L was better than the performance of Pme_H. This reversal in performance was most likely related to missing values in the spike-in proteins in the SGSDS full datasets and the inability to reliably define the more complex models related to Pme_H for all the spike-in proteins. Furthermore, the performance of the full regression modeling approach was better over a broader spectrum of trend difference categories than the performance of the lower degree models (Supplementary Fig. 1). The degree of the regression was directly associated to how complex trend differences could be detected between the conditions by the models, as expected. All polynomial regression model approaches utilized orthogonal polynomials. This likely allowed the detection of both lower and higher order differential trend patterns simultaneously by reducing collinearity between the coefficients of different polynomial degrees as compared to using raw polynomials[58,59].

Most of the examined approaches in this study require a suitable polynomial degree or degrees of freedom to be defined. However, our results show that, through the use of orthogonal polynomials, differential expression related to different polynomial degrees can be effectively detected simultaneously within a single model (Supplementary Fig. 1). Moreover, the performance of RolDE was not sensitive to the specific polynomial degrees and remained excellent regardless of their choice (Supplementary Fig. 5). By default, RolDE defines the used degrees automatically. Similar to the generalized additive

modeling (GAM) framework[60,61], where the response variable is modeled against unknown smooth functions of explanatory variables, RolDE can simultaneously search for many different types of patterns related to longitudinal differential expression. While the advantage of GAM is that it is not limited to global parametric functions, such as polynomials, which provides flexibility to adapt the fit to the data even with complex non-linear relationships, its downside is the propensity to overfit. Therefore, with RolDE we decided to focus on relatively simple models, which is crucial for typical longitudinal proteomics studies, involving small to moderate numbers of individuals and time points.

By default, the use of RolDE requires the user to provide only a design matrix linking the samples in the data to the time points and conditions, and information whether the time points are aligned or not. RolDE then detects any kind of longitudinal differential expression (e.g., stable differences in expression levels, differences in linear/quadratic/cubic/sigmoid/etc. longitudinal expression patterns) from the data. RolDE is freely available as an R package in Bioconductor (https://bioconductor.org/packages/RolDE/).

## Methods

### Semi-simulated spike-in proteomics datasets
Three spike-in proteomics datasets were used as a basis for generating the semi-simulated longitudinal spike-in proteomics datasets.

**The Universal Proteomics Standard Set (UPS1) data.** The UPS1 spike-in data includes 48 Universal Proteomics Standard Set (UPS1) proteins spiked into a yeast proteome digest with five different concentrations: 2, 4, 10, 25, and 50 fmol/µl[30]. Three technical replicates of each concentration were analyzed using an LTQ Orbitrap Velos mass spectrometer. After preprocessing, 47 spike-in proteins and 1581 proteins remained in the UPS1 data.

**The Shotgun Standard Set Dataset (SGSDS).** In the SGSDS spike-in data, 12 nonhuman proteins were spiked into a stable human background (human embryonic kidney, HEK-293) in eight different sample groups with known concentrations[31]. Three technical replicates of each sample group were analyzed using a Q Exactive Orbitrap mass spectrometer both in the DDA and data-independent acquisition (DIA) modes. In this study, the DDA shotgun proteomics data was used. After preprocessing, all 12 spike-in proteins and 3487 proteins remained in the SGSDS data.

**The Clinical Proteomic Tumor Analysis Consortium (CPTAC) dataset.** The CPTAC data from study six contains 48 UPS1 proteins spiked into a stable yeast proteome digest in five different concentrations: 0.25, 0.74, 2.2, 6.7 and 20 fmol/µl[32]. Three technical replicates of each sample group were analyzed using an LTQ Orbitrap mass spectrometer (test site 86). After preprocessing, 41 spike-in proteins and 1247 proteins remained in the CPTAC data.

The default peak-picking settings were applied to process the raw MS files of all the datasets in MaxQuant[62] version 1.5.3.30. Peptide identifications were performed using the Andromeda search engine. The 'match between the runs' option was enabled with the default time window of 0.7 min and alignment time window size of 20 min. 'Require MS/MS for comparisons' was on, and decoy mode was 'revert'. False discovery rate (FDR) of 0.01 was set as a threshold for peptide and protein identifications. A FASTA database of the yeast *Saccharomyces cerevisiae* protein sequences merged with the Sigma-Aldrich 48 UPS1 protein sequences was used to search protein identifications for the UPS1 and CPTAC data. For the SGSDS data, a FASTA database of the human HEK-293 cell proteins merged with the sequences of the nonhuman spike-in proteins was used. MaxLFQ with 'advanced ratio estimation', 'stabilize large LFQ ratios' and 'advanced site intensities' were on. Non-normalized protein intensities were extracted from MaxQuant and imported into the R statistical programming software environment.

The datasets were normalized using the variance stabilization normalization (vsn)[63] shown to perform well with proteomics spike-in data[12].

### Generation of the semi-simulated datasets with varying longitudinal trends

The normalized spike-in datasets were used to create semi-simulated datasets with varying pre-defined longitudinal trends in the spike-in proteins. Unlike the spike-in proteins, the expression of the background proteins was expected to remain stable in all sample groups. It should be noted, however, that due to experimental noise and fluctuations, there is always some variation also in the abundance of the background proteins, which reflects the nature of MS data in a real biological experimental setting.

For the creation of the semi-simulated datasets, the means and standard deviations of the proteins in the sample groups of the original normalized spike-in data were used. More precisely, the simulated expression of protein $i$ in sample group $j$ for each replicate was drawn from a normal distribution $N(\mu_{ij}, \sigma_{ij}^2)$, where the protein and sample group specific mean $\mu_{ij}$ and variance $\sigma_{ij}^2$ were calculated from the original spike-in dataset. The longitudinal trends were created by reorganizing the semi-simulated sample groups into desired combinations. For example, to create a simple linear trend using the UPS1 data, we sequentially combined the simulated 2 fmol to 50 fmol sample groups. This approach allowed the generation of a plethora of semi-simulated proteomic datasets with different longitudinal trends for the spike-in proteins as well as a constant but noisy background and, therefore, a realistic benchmarking of the longitudinal methods.

Six basic trend categories were introduced for the UPS1 and CPTAC datasets: Stable, Linear, LogLike, Sigmoid, Poly2, and PolyHigher (Supplementary Data 1). For the SGSDS datasets, five basic trend categories were used: Stable, LogLike, Sigmoid, Poly2, and PolyHigher. For the SGSDS datasets, a longitudinal linear increase or decrease (Linear) was unattainable due to the uneven concentration differences of the spike-in proteins between the samples in the original data[31]. In total, this approach resulted in 21 trend difference combinations between two simulated conditions for the UPS1 and CPTAC datasets and 15 combinations for the SGSDS dataset. When each dataset contained only one type of trend for a condition, a total of 300 semi-simulated datasets with two conditions and different longitudinal trends and/or expression levels were generated for the UPS1 and CPTAC datasets and 210 for the SGSDS dataset (Supplementary Data 1).

In addition, we generated 300 semi-simulated datasets with varying trends within a condition and varying trend differences between the conditions to reflect typical real experimental data where proteins with multiple different types of longitudinal trends coexist. These datasets were generated by randomly selecting 10 different trend differences for the spike-in proteins using the UPS1 dataset, referred to as UPS1 Mix in the results section (Supplementary Data 1).

For benchmarking the methods, two versions of each semi-simulated UPS1 and SGSDS dataset were generated, referred to as full and filtered in the results section. In the full datasets, none of the proteins were filtered from the data (i.e., proteins with missing values were included), whereas the filtered datasets contained only proteins without any missing values.

As missing values are a common occurrence in proteomics data[17,18], the CPTAC dataset with relatively large numbers of missing values in the spike-in proteins was used to generate semi-simulated datasets with high proportions of missing values to explore the tolerance of the different methods against such missing values. Here, only full datasets were considered.

### Longitudinal *Francisella tularensis* proteomics data

To evaluate the longitudinal differential expression methods in a real proteomics study setting, we generated an experimental membrane enriched longitudinal proteomics data of *Francisella tularensis* subspecies *novicida (Fn)*. *Francisella tularensis* subspecies *tularensis (Ft)* is a highly pathogenic Gram-negative bacterial agent responsible for the disease tularemia in humans. Research interest on *Ft* has peaked during the past decades due to the possible application of the bacterial agent in biological warfare and bioterrorism[64]. As reliable methods for the genetic manipulation of the highly virulent subspecies *Ft* have been lacking, the more responsive and avirulent subspecies *Fn* was used as a proxy in an effort to shed light into the mechanisms of pathogenicity of *Ft*[64]. We considered two temperature regulated N-acyltransferases, designated lpxD1 and lpxD2, which have been identified in *Fn*. LpxD is involved in the production of a key virulence factor, lipid A, in *Fn*[50]. The null mutant of lpxD1 has been shown to be more sensitive to environmental factors and attenuated in virulence when compared to the wild type[50,51]. In addition, we studied a late acyltransferase lpxL, which has been discovered in *Ft* and shown to be essential for cell viability at temperatures above 33 °C[65]. Thus, the dataset consisted of four strains: the wild type (WT) and null mutants of lpxD1 (D1), lpxD2 (D2), and lpxL (L). In order to activate the temperature sensitive enzymes responsible for the production of the lipopolysaccharide, the global protein expression of the *Fn* strains was examined at five temperatures: 18, 21, 25, 32, and 37 °C (surrogates for time points). Given that LPS is assembled in the membrane of Gram-negative bacteria, we chose to isolate and generate data only for a membrane-enriched fraction. Each mutant and wild type strain consisted of three biological replicates (surrogates for individuals), and each biological replicate was measured in three technical replicates, resulting in a total of 36 unique samples at each temperature and 180 samples in total. For evaluation of the methods, differential expression between the null mutant and the wild type strains was investigated, while their growth at different temperatures offered a surrogate for the different time points.

Wild-type *Fn* strain U112 was originally obtained from Professor Francis Nano (University of Victoria, Canada)[66] and maintained at −80 °C. The lpxD1, lpxD2, and lpxL null mutants were generated previously[67]. The lpxL strain used was a transposon insertion mutant of FTN_0071 (*lpxL*) identified as tnfn1_pw060510p04q127 carrying a kanamycin cassette disrupting the coding sequence. The mutant was functionally characterized as having an lpxL knockout lipid A phenotype. All bacterial strains were grown in tryptic soy broth supplemented with 0.1% cysteine at the designated temperatures (18, 21, 25, 32, and 37 °C). Cultures were harvested at log phase, supernatant was aspirated, and cell pellets were flash frozen and stored at −80 °C until processing.

To isolate a fraction of proteins enriched in membrane proteins, cell pellets were first fractionated along the lines of a previous study[68]. Briefly, cell pellets were resuspended in 0.1 M NaPO$_4$, 0.05 M MgSOD4, DNaseI digested, and sonicated. Unbroken cells were removed by centrifugation and the supernatant was again subjected to centrifugation at 39,000 × $g$ for 45 min. The resulting pellet, which contained the membrane envelope, was resuspended in 50 mM ammonium bicarbonate and subjected to trypsin digestion. After digestion, samples were desalted using MACROspin C18 columns (The Nest Group, Southborough, MA). The flow-through was collected, concentrated in a speedvac to near dryness, and resuspended in 5% acetonitrile/0.1% formic acid for MS analysis. Samples were stored at −80 °C until MS analysis.

The membrane samples were prepared for proteomics analysis by high performance liquid chromatography (HPLC)-tandem mass spectrometry on an LTQ Orbitrap Elite (Thermo-Fisher) according to previously published protocols[69]. Briefly, mass spectrometry data were collected over a 95 min LC gradient using data-dependent acquisition with a full MS scan for 350–2000 *m/z* range at 120 K resolution. Top 15 ions were selected with an isolation width of 2 *m/z* for fragmentation by CID, with a dynamic exclusion of 30 s. The default peak-picking settings were applied to process the raw MS data files in MaxQuant[62] version 1.6.5.0. Peptide identifications were performed using the

Andromeda search engine with a SwissProt/UniProt FASTA database of all reviewed and unreviewed protein sequences for *Fn* strain U112 (April 2019). Trypsin digestion with a maximum of two missed cleavages, carbamidomethylation of cysteine as a fixed modification, and methionine oxidation and N-terminal acetylation as variable modifications were used as search parameters. Precursor mass tolerance was set to 20 ppm and fragment mass tolerance to 0.5 Da. Minimum peptide length was set to 7. The 'match between the runs' option was enabled with the default time window of 0.7 min and the alignment time window size of 20 min. 'Require MS/MS for comparisons' was on, and decoy mode was 'revert'. A false discovery rate (FDR) of 0.01 at the peptide and protein level was applied. The MaxQuant label-free quantification LFQ algorithm was used to calculate the relative protein intensity profiles across the samples. 'Advanced ratio estimation', 'stabilize large LFQ ratios' and 'advanced site intensities' were on. Non-normalized protein intensities were extracted from MaxQuant and imported into the R statistical programming software environment version 3.6.1. Reverse protein hits, known contaminants, and proteins with less than two peptides and at least one unique peptide were filtered out. The dataset was normalized using the variance stabilization normalization (vsn)[63] shown to perform well with proteomics data[12].

## Longitudinal human iTreg proteomics data

To further evaluate the methods on real experimental biological data, we included a previously published proteomics dataset exploring the molecular landscapes associated with in vitro induction of human iTreg cells[33]. Naive CD4$^+$ T cells from three adult donors were stimulated and iTreg cells were induced using two different protocols with or without the vitamin A metabolite all-trans retinoic acid (ATRA) and rapamycin (groups G03 and G05 in the original study). Samples were collected at 6 h, 24 h, 48 h and 6 d of differentiation and subjected to proteomics analysis. The samples were analyzed using TMT-10plex-based LC-MS/MS with a Q Exactive Hybrid Quadrupole-Orbitrap mass spectrometer[33] and the data was normalized using the TMT ratio sample median normalization. The preprocessed data available from the original study was downloaded and log$_2$-transformed. Longitudinal differential expression was determined between the iTreg and the mock-stimulated control cells (group G02 in the original study), separately for both protocols, resulting in two comparisons.

## Longitudinal human type 1 diabetes proteomics data

Finally, we evaluated the performance of RolDE on human type 1 diabetes proteomics data from a recent study by Liu et al.[10]. The data contained longitudinal blood plasma protein expression measurements from 11 children (aged 0.8–14.4 years) developing T1D and 10 matched controls. Nine longitudinal samples were collected for each child, with the T1D case samples covering seroconversion and clinical T1D diagnosis. The samples were analyzed using a TMT-10plex-based LC-MS/MS approach with a Q Exactive HF mass spectrometer with all time points of one individual and a common pooled reference sample included in one TMT-10plex MS run[10]. Altogether, 189 samples were analyzed. The data was normalized and filtered similarly to the original study: the reporter ion intensities were standardized to the reference sample, log$_2$-transformed and median normalized, and only proteins with at least two individuals from either group with at least seven non-missing values were included in the analysis[10].

## Robust longitudinal Differential Expression (RolDE)

The proposed method, RolDE, is a composite method, consisting of three independent modules with different approaches to detect longitudinal differential expression (see Fig. 1e for a schematic illustration). While each module typically performs well already on its own in ranking the true findings high, there can also be some additional unwanted noise detections (false positives) associated with each module. Since such false positives are typically specific for a single module, combining the modules creates a balanced composite method, enriches the true signal at the top of the results and allows RolDE to robustly detect varying differences in longitudinal trends and expression levels in diverse data types and experimental settings. The three modules of RolDE, named RegROTS, DiffROTS and PolyReg, and their combination are described in more detail below.

## RegROTS

The RegROTS module combines the power of polynomial regression modeling and the reproducibility optimization of ROTS[26]. First, a polynomial regression model for each protein is separately fitted for each individual $u$ over time $t$:

$$y_u = \beta_{0u} + \sum_{j=1}^{d} \beta_{ju} t^j + \varepsilon \tag{1}$$

where $d$ is the polynomial degree of the model, $\beta_{ju}$ are the regression coefficients, and $\varepsilon$ is the error term. The degree $d$ is by default defined as: $d = \max\{1, \min(\lfloor \frac{m}{2} \rfloor, 4)\}$, where $m$ is the median number of time points over all the individuals. Orthogonal polynomials are used to reduce multicollinearity and allow for more independent exploration of coefficients of different polynomials of the same variable[58,59]. The orthogonal polynomials are defined using the three-term recursion algorithm described in[70] and implemented in the function poly of R. If the time points in the compared conditions are aligned, we use the default algorithm. If the time points are not aligned, we omit the last scaling step in the function for each polynomial (i.e., the square root of the sum of the squared vector values of the longitudinal variable, the L2 norm) to preserve comparability between the coefficients of the different individual regression models.

Next, all coefficients of the same degree $j$ are compared across the individual models by considering all possible pairs of individuals $u$ and $v$ between the two conditions $C_1$ and $C_2$: $\Delta\beta_{juv} = \beta_{ju} - \beta_{jv}$, where $u \in C_1$ and $v \in C_2$. These replicate comparisons are divided into multiple different runs so that each individual is used at most once in each run. Thus, assuming an equal number of individuals per condition $n$, a total of $n$ runs are considered. For instance, with three individuals in both conditions, we consider three runs: run 1 $u_1$-$v_1$, $u_2$-$v_2$, $u_3$-$v_3$; run 2 $u_1$-$v_2$, $u_2$-$v_3$, $u_3$-$v_1$; and run 3 $u_1$-$v_3$, $u_2$-$v_1$, $u_3$-$v_2$. Subsequently, within each run, multigroup ROTS is used to test the null hypothesis $\Delta\beta_0 = \Delta\beta_1 = \ldots = \Delta\beta_d = 0$ for each protein to investigate, whether there are differences in any of the regression coefficients of the individual models between the different conditions, including the intercept $\Delta\beta_0$ (mean expression), linear longitudinal trends $\Delta\beta_1$ (slope), quadratic trends $\Delta\beta_2$, or any of the examined polynomial trends until degree $d$ ($\Delta\beta_d$). In each run, the proteins are then ranked according to the significance values of the modified *F*-statistic from the multigroup ROTS. The protein with the smallest significance value gets the smallest rank. The final RegROTS score for a protein is calculated by combining these ranks from the different runs using the rank product (i.e., the geometric mean):

$$S_{\text{RegROTS}} = \left( \prod_{i=1}^{n} r_i \right)^{\frac{1}{n}} \tag{2}$$

where $r_i$ is the rank of the protein in run $i$ and $n$ is the total number of runs. Previous cross-sectional studies have suggested the rank product approach to be especially useful for noisy data[52,53]. The empirical distribution of $S_{\text{RegROTS}}$ under the null hypothesis is shown in Supplementary Fig. 6a. Given the null hypothesis is valid, the differences in the estimated coefficients for the longitudinal trends

and expression levels between the conditions are effectively zero and no longitudinal differential expression exists.

## DiffROTS

The DiffROTS module examines directly the expression differences between the conditions at each time point. Similar to the RegROTS module, all the individuals in condition $C_1$ are compared to all the individuals in condition $C_2$ in multiple runs, but instead of comparing the coefficients of the fitted models, the expression measurements $y_{tu}$ and $y_{tv}$ for each pair of individuals $u$ and $v$ at time point $t$ are used directly: $\Delta y_{tuv} = y_{tu} - y_{tv}$. If any of the protein abundance values are missing, this results in a missing value for that time point in the analysis.

Similar to the RegROTS module, multigroup ROTS is then used within each run to detect differences in the expression levels between the conditions, with the null hypothesis that there are no differences in the expression values at the distinct time points between the conditions, i.e., $\Delta y_1 = \Delta y_2 = \ldots = \Delta y_T = 0$, where $T$ is the total number of time points considered. In each run, the proteins are ranked according to the significance values of the modified $F$-statistic from the multigroup ROTS. The protein with the smallest significance value gets the smallest rank. The final DiffROTS score for a protein is then calculated by combining these ranks from the different runs using the rank product (i.e., the geometric mean):

$$S_{\text{DiffROTS}} = \left( \prod_{i=1}^{n} r_i \right)^{\frac{1}{n}} \tag{3}$$

where $r_i$ is the rank of the protein in run $i$ and $n$ is the number of runs. The empirical distribution of $S_{\text{DiffROTS}}$ under the null hypothesis is shown in Supplementary Fig. 6b. Given the null hypothesis is valid, the expression differences between the conditions are effectively zero at each time point and no longitudinal differential expression exists.

If the time points in the data are not aligned between the individuals and the expression levels between the different conditions cannot be directly compared at different time points, the DiffROTS module is adjusted to such a design. This is done by examining the expression level differences between the conditions after accounting for time-associated trends of varying complexity in the data. More specifically, polynomial regression models of degrees $j = 0,\ldots,d$ are first fit to each protein: $y = \beta_0 + \beta_j t^j + \delta_{0u} + \varepsilon$, where $\delta_{0u}$ is the individual specific baseline, $\delta_{0u} \sim N(0, \sigma_{0u}^2)$. The maximum degree $d$ is by default defined as $d = \max\{2, \min(m-1, 5)\}$, where $m$ is the median number of time points over all the individuals. Individual variation is taken into account by adding a random effect for the individual baseline but can be adjusted to incorporate individual slopes as well. The model residuals $e_{jtu} = y_{tu} - \hat{y}_{jtu}$ are then recorded for each individual $u$ in each of their measured time point $t$, and differentially expressed proteins between the two conditions are determined by comparing all the residuals of the individuals in one condition to those in the other condition using two-group ROTS[26], separately for the different polynomial degrees $j$. To detect any differential expression between the conditions, the final DiffROTS score for a protein is determined as the minimum over the significance values $p_j$ over the comparisons: $S_{\text{DiffROTS}} = \min(p_0, p_1, \ldots, p_d)$.

## PolyReg

The PolyReg module applies polynomial regression modeling to each protein to detect longitudinal differential expression over time $t$ across the conditions $c$:

$$y = \beta_0 + \sum_{j=1}^{d} \beta_j t^j + \gamma_o c + \sum_{j=1}^{d} \gamma_j c \cdot t^j + \varepsilon \tag{4}$$

where $d$ is the polynomial degree of the model, $\beta_j$ and $\gamma_j$ are the time and condition-related regression coefficients, respectively, and $\varepsilon$ is the error term. Individual variation can be taken into account by adding a random effect for the individual baseline or slope. By default, RolDE uses fixed effects models for the PolyReg module when the time points are aligned and mixed models with random effects for the individual baseline when the time points are not aligned. Using the mixed effects modeling approach should be considered especially when heterogeneity and uneven sampling points (e.g., non-aligned time points) are expected among individuals. The degree $d$ is by default defined as $d = \max\{2, \min(m-1, 5)\}$, where $m$ is the median number of time points over all the individuals. For each of the condition related coefficients $\gamma_j$, the null hypothesis is $\gamma_j = 0$, that is, the condition related variable is not statistically significantly associated with the abundance of the protein. Each null hypothesis is examined using $t$-test, which is the standard way of assessing the significance of regression coefficients. The corresponding significance values $p_{\gamma_j}$ are then used to determine the final score for the PolyReg module, $S_{\text{PolyReg}}$, which is calculated as the minimum over the significance values of the condition related regression coefficients:

$$S_{\text{PolyReg}} = \min(p_{\gamma_0}, p_{\gamma_1}, \ldots, p_{\gamma_d}) \tag{5}$$

## The composite method RolDE

Finally, for a comprehensive inspection of differential expression between the conditions, the results from the different modules are combined using the rank product (i.e., geometric mean of the ranks) of the module scores:

$$S_{\text{RolDE}} = \sqrt[3]{r\left(S_{\text{RegROTS}}\right) \cdot r(S_{\text{DiffROTS}}) \cdot r\left(S_{\text{PolyReg}}\right)} \tag{6}$$

To estimate the significance of the RolDE score, a simulation procedure is used. Given the null hypothesis is true, the significance value distribution within a RegROTS or DiffROTS run is (approximately) uniform. Based on this, a simulated internal rank product is calculated for the RegROTS and DiffROTS modules as follows. First, an equal number of simulated significance values as there are experimental significance values in the corresponding run are generated from the uniform distribution. Second, the simulated significance values within each run are ordered so that the ranks of the proteins according to the experimental significance values are retained to account for the dependencies between runs. Finally, each simulated significance value within each run is replaced with the rank of the closest experimental significance value and these are used to calculate the simulated internal rank products. Given the null hypothesis is valid, the experimental and simulated internal rank products are similar (Supplementary Fig. 6c). For the PolyReg module, the representative simulated significance values are calculated in a similar fashion. Under the null hypothesis, the significance value distribution of the condition related coefficients is uniform. First, an equal number of simulated significance values as there are experimental significance values are generated from the uniform distribution. Second, the simulated significance values are ordered according to the overall order of the experimental significance values to account for any potential dependencies between the coefficients of different polynomial degrees for a protein. Finally, representative significance values for proteins are calculated as the minimum over all the simulated significance values. Given the null hypothesis is valid, the experimental and simulated significance values are similar.

After acquiring the simulated internal rank products or representative significance values for each module, final simulated rank products are calculated equivalently to the experimental rank products and utilized to estimate the overall significance values. This is

done by generating 500,000 (by default) simulated rank products and then calculating the fraction of simulated rank products smaller or equal to each experimental rank product. Given the null hypothesis is valid, the estimated significance value distribution for RolDE will be approximately uniform (Supplementary Fig. 6d). By default, the estimated significance values are adjusted for multiple hypothesis testing using the Benjamini-Hochberg procedure[71]. Alternatively, Bonferroni correction, Q-value adjustment or any other multiple testing correction method can be used.

The RolDE workflow is demonstrated in Supplementary Fig. 7. For all the semi-simulated spike-in datasets, and the *Fn* and iTreg data with technical replicates or biological cell culture replicates, RolDE with fixed effects regression was applied. For the T1D data with human individuals, RolDE with mixed effects regression with random effects for the individual baseline was applied.

In Supplementary Note, we provide a refined evaluation of the proposed RolDE methodology, including the sensitivity of the method to various adjustable parameters (but not needed to be adjusted by default, Supplementary Fig. 5), the effect of the reproducibility-optimization with ROTS (Supplementary Fig. 8) and the combination of multiple rankings on the performance (Supplementary Fig. 9), as well as the effectiveness of the simulation approach in estimating the significance values and controlling the number of false discoveries (Supplementary Table 3).

## Existing methods for detecting longitudinal differential expression tested

**Reproducibility Optimized Test Statistic (BaselineROTS).** ROTS is a well-established differential expression method aiming to maximize the reproducibility of the top detections using a modified *t*-statistic and group preserving bootstraps[26]. It has been observed to perform well in multiple types of omics data, including proteomics[12,30,54,55]. In this study, ROTS was used as a baseline cross-sectional method at each time point against which the longitudinal methods were compared to. To detect any differences in longitudinal expression, the minimum significance value over all the time point comparisons was recorded as the representative significance value for each protein. Version ≥ 1.12.0 of the Bioconductor R package ROTS was used in this study.

**Bayesian estimation of temporal regulation (BETR).** BETR utilizes the Bayesian framework and was first introduced for the analysis of times series DNA microarray data[13]. It calculates the probability of differential expression for each feature (e.g., gene or protein), taking correlations between time points into account; the magnitude of expression at time points closer to each other are assumed to be more correlated than those further apart[13]. Version ≥ 1.32.0 of the Bioconductor R package betr was used in this study.

**Linear models for microarray data (Limma, LimmaSplines).** Limma is a popular toolset used especially in the analysis of gene expression microarray and RNA-seq data. Longitudinal differential expression can be examined in two different ways according to the Limma user's guide[16]. The first option is through group-mean parametrization, where the design matrix for the linear models is designed is such a way that a separate coefficient exists for each time point and condition combination and various contrasts between the coefficients can be defined to examine differential expression between the conditions. The contrasts were defined as differences in expression between the conditions at each time point and differential expression between the conditions was determined by examining whether all contrasts were zero simultaneously using the moderated *F*-statistic. The second option included using natural cubic spline curves to model the longitudinal variable over time and fitting separate curves for the conditions for each protein from which differences related to the conditions

were explored, following the instructions in the limma R package manual[16]. Version ≥ 3.40.2 of the Bioconductor limma R package was used in this study.

**Timecourse.** Timecourse ranks features (e.g., genes or proteins) according to probabilities for differential expression using the Maxwell–Boltzmann or the Hotelling's $T^2$ statistics through a multivariate empirical Bayes approach, taking replicate variances and correlations in expression levels between time points into account[14]. The method borrows information across features to better estimate the variance-covariance matrices to reduce the number of false positives and false negatives[14]. Version ≥ 1.56.0 of the Bioconductor timecourse R package was used in this study.

**Microarray significant profiles (MaSigPro).** MaSigPro is a method originally developed for the analysis of time series gene expression microarray data[15]. It follows a two-step regression approach to detect longitudinal differential expression between the conditions[15]. In the first step, a general regression model is defined for each feature (e.g., gene or protein). In the second step, only the significant models from the first step are then modeled using polynomial regression to find differences between the compared conditions[15]. Version ≥ 1.56.0 of the Bioconductor maSigPro R package was used in this study. For each protein, a minimum over the condition-related significance values was used as a representative significance value.

**Linear mixed effects regression modeling (Lme).** Linear mixed effects regression is a popular approach in modeling longitudinal data, where individual variability can be incorporated into the model in the form of random effects[10]. We considered two different types of models. In the first variant, a random effect only for the individual baseline (the intercept) was allowed. In the second variant, a random effect was allowed also for the slope. Finally, a likelihood ratio test was conducted to examine if adding a random effect for the slope yielded a significantly better fit at the significance level of 0.05. If this was not the case or the second model could not be defined due to insufficient number of measurements (caused e.g., by missing values), the model with a random effect only for the intercept was used for the protein. For each protein, a minimum over the condition-related significance values was used as a representative significance value. The R package nlme version ≥ 3.1-142 was used for all the mixed effects modeling approaches in this study.

**Polynomial mixed effects modeling (Pme).** In the polynomial mixed effects approach, we added polynomial fixed terms for the time-related effects in the models. Again, two different types of models were considered: models with a random effect only for the individual intercept and models with a random effect also for the slope. Similarly as with Lme, a likelihood ratio test was conducted to examine if adding a random effect for the slope yielded a significantly better fit at the significance level of 0.05. We did not consider random effects for the higher order polynomials as there was typically not enough data in the relatively short time series to define such models. For each protein, a minimum over the condition-related significance values was used as a representative significance value.

**Extraction of differential gene expression (EDGE).** EDGE[27] is a well-established method developed for identifying differentially expressed genes in time course studies with DNA microarrays. It uses regression splines to fit two models for each feature: a null model (no longitudinal differential expression between the conditions) and the full model (differential expression between the conditions). A likelihood ratio test (LRT) or an optimal discovery procedure (ODP) can be used to determine whether the full model fits the data better than the null model, i.e., whether there is longitudinal differential expression. The

LRT approach of the Bioconductor R package edge version ≥ 2.16.0 was used in this study.

**Linear mixed model spline framework for analysing time course omics data (LMMS).** The LMMS[28] approach utilizes linear mixed effects model regression splines to detect longitudinal differential expression. Fitted models for longitudinal differential expression between the conditions are compared to the null model using likelihood ratio tests. The default cubic regression splines in the R package lmms version ≥ 1.3.3 were used. For each protein, a minimum over the condition-related significance values was used as a representative significance value.

**Omics longitudinal differential analysis (OmicsLonDA).** OmicsLonDA[29] offers a statistical framework for identification of time intervals with significant differences between the conditions. First, the measurements for each feature are adjusted according to the baseline for each individual. Regression splines are then fitted for the features and Monte Carlo permutations are used to generate empirical distributions for the test statistic under the null hypothesis. The Bioconductor R package OmicsLonDA version ≥ 1.6.0 was used. For each protein, all possible time intervals were explored and a maximum value for the test statistic over all the explored intervals was used as a representative value to detect any longitudinal differential expression.

For each polynomial regression-based approach (LimmaSplines, MaSigPro, Pme, EDGE), we explored two different levels of complexity: a more complex model (denoted by the extension H in the results) with the degree of the polynomial set to $T-1$, and a less complex one (denoted by the extension L) with the degree set to $\lfloor T/2 \rfloor$, where $T$ is the number of time points.

R statistical programming software environment ≥ 3.6.1 was used for the computational analysis performed in this study.

## Evaluation of the methods in the semi-simulated spike-in proteomics datasets

In the semi-simulated datasets, we evaluated the performance of the longitudinal methods in their ability to correctly detect true differential expression (the spike-in proteins) using receiver operating characteristic (ROC) analysis. In the ROC analysis, the sensitivity (i.e., true positive rate) was plotted against the specificity (i.e., true negative rate). The area under the ROC curve (AUC) was used to measure how well a given method was able to distinguish the true signal of interest when the detection threshold (e.g., significance value) was varied. As typically the interest of a differential expression analysis is focused on the top findings, we used partial AUC (pAUC) between specificity values 1 and 0.9 to score the methods on the essential part of the ROC curve. The calculated pAUCs were summarized using the interquartile range (IQR) mean across the datasets of each type. The R package pROC version ≥ 1.15.3 was used to conduct the ROC analysis[72]. The pAUCs were visualized using the vioplot R package version ≥ 0.3.2.

Different methods had varying abilities to calculate a score (a test statistic or a ranking) for the examined proteins due to missing values or other reasons. To ensure comparability, a full result list was expected from each method, including a score for all the proteins in the examined dataset. If a method could not produce a score for a protein, a random score larger than the maximum observed score for the method in the dataset was generated, placing the protein at the end of the result list. Thus, all proteins without valid scores for a given method, were placed randomly at the end of the result list.

## Reproducibility and biological relevance in the longitudinal *Fn* data

In the experimental longitudinal *Fn* dataset, we assessed the reproducibility of the results of the methods using the three technical replicate datasets. After filtering out all proteins with missing values,

three completely separate technical datasets were formed, which were assumed to be similar to each other. Longitudinal differential expression between all possible pairwise combinations of strains (WT vs. D1, WT vs. D2, WT vs. L, D1 vs. D2, D1 vs. L, D2 vs. L) were considered. To estimate the overall reproducibility of each method, the similarity of their outputs in the replicate datasets was assessed using the Spearman's rank correlation coefficient. To evaluate the reproducibility of the top differential expression findings, the median proportional overlap between the top *k* findings was calculated over the replicate datasets, when the examined top list size *k* was varied from 1 to the number of proteins in the complete dataset. The proportional overlap at each value of *k* was calculated as the median overlap over the replicate datasets divided by *k*.

To examine the biological relevance of the findings by each method in the *Fn* dataset, we examined how the proteins in the KEGG pathway Lipopolysaccharide biosynthesis−*Francisella tularensis* subsp. *novicida* U112 (ftn00540) complemented with the proteins in the associated Lipopolysaccharide biosynthesis knockout pathway (ko00540) (Supplementary Table 1) were ranked by the different methods. One of the key virulence factors in *Fn* is lipid A, which is part of LPS, whereas LPS is an essential part of the Gram-negative bacterial outer membrane and forms an amphipathic interface between the bacteria and the environment[49,50]. As the modified null mutants of acyltransferases in the *Fn* data (D1, D2, L) are associated with lipid A and LPS, the 18 proteins in the selected ftn00540 and ko00540 pathways are assumed to be affected by the modifications. To explore this, we used Gene Set Enrichment Analysis (GSEA)[35] to investigate enrichment of the relevant pathway proteins in each wild type to mutant strain comparison in the normalized data with technical replicates averaged for each biological replicate. For a meaningful and fair comparison, the relevant and included pathway proteins in each comparison were determined as those pathway proteins with enough variation to be detected by the methods by having a coefficient of variation (CV) larger than the median CV for all the proteins (Supplementary Table 1). The result lists for each method in each comparison were ranked according to the strength of differential expression and used for preranked GSEA. Again, if a method failed to provide a score for a protein (due to missing values or other reasons), a random score larger than the maximum observed score for the method in the comparison were generated, placing these proteins at the end of the result list. The gene set analysis was performed using the Bioconductor R package fgsea version ≥ 1.10.1. The normalized Enrichment Scores (NES) were used to reflect the ability of the methods in detecting the relevant pathway proteins and in producing biologically meaningful results.

## Biological relevance and shared findings in the longitudinal human iTreg data

To further evaluate the biological relevance of the results by the different methods, we explored how the methods detected proteins in known Treg-related gene sets in the human iTreg data. Altogether 28 human Treg related gene sets were downloaded from the Molecular Signatures Database (MSigDB) version 7.5.1[35] (Supplementary Data 2). These gene sets included up- and downregulated signature genes in Treg states from altogether 14 different scenarios. For our purposes, each pair of up- and downregulated gene sets were combined into a single gene set, resulting in 14 gene sets from MSigDB in total. In addition, three additional gene sets were downloaded. The first set included genes with a mean fold change ≥ 2 in Treg cells as compared to conventional CD4$^+$ T cells from a large human study including 168 donors[34]. As Treg cells are known to express the transcription factor forkhead box P3 (FOXP3)[33], the associated Reactome pathway (RUNX1 and FOXP3 control the development of regulatory T lymphocytes) was included as a second additional gene set. Finally, interactome of FOX3P from the STRING protein-protein interaction database was downloaded, including 100 known and predicted interacting proteins for

FOXP3. All the used gene sets are given in Supplementary Data 2. Functional enrichment for each gene set and each method was determined using GSEA similarly as with the *Fn* dataset.

To evaluate the performance of the methods in terms of unique and shared findings in the iTreg data, the result lists for each method were ranked according to the strength of the differential expression. The overlap of the top 1000 findings was investigated. For each of the top 1000 findings for each method, the number of other methods sharing the same protein in their respective top 1000 findings was determined. For a fair evaluation of methods with multiple variants, only the best-performing variant was included, as the different variants share a considerable proportion of their result lists. The selected variants included the higher order variants for Pme, MaSigPro and EDGE, and the group-mean parametrization approach for Limma.

### Reporting summary

Further information on research design is available in the Nature Portfolio Reporting Summary linked to this article.

## Data availability

The UPS1 spike-in dataset is available from the PRoteomics IDentification Database (PRIDE) with the identifier PXD002099. The SGSDS spike-in dataset is available from PeptideAtlas: No. PASS00589. The CPTAC spike-in dataset (study 6, at test site 86) is available from the CPTAC Portal (https://cptac-data-portal.georgetown.edu/cptac/study/list?scope=Phase+I).

The *Francisella tularensis* subspecies *novicida* (*Fn*) data generated in this study has been deposited in the ProteomeXchange Consortium via the PRIDE partner repository under accession code PXD025439.

The longitudinal type 1 diabetes proteomics dataset of Liu et al. is available from the original publication[10]. For this study, the data available in Table S3 and the clinical data available in Table S1 and Table S2 of the supplementary material from the original publication were downloaded and used. The longitudinal iTreg proteomics dataset of Schmidt et al. is available from the original publication[33]. For this study, the data available in the Additional file 3: Table S2 from the original publication was downloaded and used. Source data are provided with this paper.

## Code availability

RolDE is freely available as an R package in Bioconductor at https://doi.org/10.18129/B9.bioc.RolDE. Custom codes for methods used in the study are available from GitHub (https://github.com/elolab/RolDE-benchmarking).

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

## Acknowledgements

The authors acknowledge support from the European Research Council ERC (677943, L.L.E.), Academy of Finland (296801, 310561, 314443, 329278, 335434, 335611 and 341342, L.L.E.), Sigrid Juselius Foundation (L.L.E.), the University of Turku Graduate School Doctoral Programme in Technology (T.V.), the National Institutes of Health (R01GM111066 and 1R01AI147314-01A1, R.K.E. and D.R.G.), and the International Centre for Cancer Vaccine Science project carried out within the International Research Agendas program of the Foundation for Polish Science co-financed by the European Union under the European Regional Development Fund (MAB/2017/03, D.R.G.). We also thank ELIXIR Finland for computational resources and Esko Pakarinen for technical support.

## Author contributions

T.V. planned the computational experiments, developed the RolDE method, performed the computational analysis, interpreted the results, and drafted the manuscript. T.S. planned the computational experiments, participated in the computational analysis and method development, interpreted the results, prepared the figures, and participated in writing the manuscript. C.E.C. planned the experiments, generated the experimental *Fn* data, and participated in analyzing the *Fn* data and writing the manuscript. A.J.S. constructed the mutants, planned the experiments for the *Fn* data, and interpreted the Fn data. B.Q.T. was responsible for the experimental design and acquisition of the *Fn* data. R.K.E. planned the experiments for the *Fn* data, interpreted the *Fn* data, and participated in writing the manuscript. D.R.G. planned the experiments for the *Fn* data, interpreted the *Fn* data, and participated in writing the manuscript. L.L.E. conceived, designed and supervised the study, participated in the method development, interpreted the results, and participated in writing the manuscript.

## Competing interests

The authors declare no competing interests.
