## [Peer Review File · Nature Communications]

REVIEWER COMMENTS

Reviewer #1 (Remarks to the Author):

This manuscript, titled 'Enhanced longitudinal differential expression detection in proteomics with robust reproducibility optimization regression', proposes that RoIDE, as a robust method applicable to noisy longitudinal proteomics data, combines the detection power of reproducibility optimization and longitudinal regression modeling. The authors clearly show an extensive evaluation of the proposed methods' performance against many existing longitudinal approaches using semi-simulated proteomics datasets with diverse scenarios as well as real experimental biological data, even with non-aligned time points. Also, this proposed method is implemented as an R package in Github and publicly available.

Main comments:

- As the authors address, there are many methods available developed for gene expression microarray data. The authors addressed that these methods do not take into account the characteristics of MS proteomics data. What characteristics of MS proteomics data are not taken into account for these methods? In the introduction, 'missing values' were mentioned as an example. Then, please explain why RoIDE is tolerant for missing values methodologically. For example, RoIDE has a better performance than Timecourse for SGSDS full, which has more missing values. Timecourse performs similarly as RoIDE on other datasets though. The methodological justification for how RoIDE overcome the missing value issue is missing in this version of the manuscript. Also, what other characteristics of MS proteomics data are taken into account for RoIDE?
- Please comment about the experiment with less than 5 time points. As the authors mentioned in the introduction, 3 or 4 time points are typical for a biomedical study. The datasets tested in the manuscript have 5, 8, or 9 time points. Can you show the performance for the cases with less than 5 time points or at least comment it?

Minor comments:

- The supplementary figure numbers seem not matched in the main text. Need to double-check them : For example, 1) page 7, line 44 : suppl fig 2 ->3, 2) Page 15, line 33 : suppl fig 4A -> 1A 3) Page 16, line 5 : suppl fig 4B -> 1B?
- Please clarify what page 10, line 27, 'what kinds of differences are searched from the data' means.

- Suggest submitting RoIDE R package in GitHub to Bioconductor. It manages all dependencies and reviews the codes for the R package, which supports the long-lifetime for R packages and usability.

Reviewer #2 (Remarks to the Author):

The manuscript describes a statistical methodology to find differentially expressed proteins in longitudinal proteomics analysis. It is an important and timely subject, for which new results are needed in the proteomics community.

The proposed approach is based on interesting rationale and show interesting preliminary results on benchmarks datasets. However, in its current state, the manuscript failed to convince me. As is, the article is focused on the results obtained on the benchmark datasets, as if they were discoveries of interest, while their only purpose is to support a new method, which (too) short presentation is postponed to the method section (together with the MS routines and state of the art methods), and which exhaustive evaluation is missing. Hereafter follows a more detailed description of my concerns about (i) the insufficient description; (ii) the insufficient justification; (iii) the biased comparative evaluation; (iv) the scope of application of the proposed methodology.

First, the method is not sufficiently described:

* RegROTS step:

- What family of orthogonal polynomials are used in RegROTS?

- How does one switches from the $\delta\beta$'s to the R_i 's of the unnumbered equation line 28 page 14?

- In the text above the equation, it is indicated that the significance value of the R_i 's are used, while the equation directly refers to the R_i 's, not their significance values. This needs clarification.

- More generally, the entire paragraph above this equation only provides a hand-waving description, or an example that is not sufficient to thoroughly understand the methodology or to re-implement it.

- What is the empirical distribution of S_{RegROTS} under the null hypothesis?

- What is the concrete meaning of the proposed null hypothesis? (i.e., the equality of the $\delta\beta$'s to zero)

* DiffROTS step:

- How missing values are dealt with? It looks like they are simply considered similarly as unaligned values, so that they are replaced by regression estimates... This must be clarified.
- What is the polynomial regression at use? (RegROTS or PolyReg one) why?
- Same ambiguity as with RegROTS about R_i 's or their significance values.
- What is the empirical distribution of S_{DiffROTS} under the null hypothesis?
- To which extent the null hypothesis from DiffROTS differs from that of RegROTS? Is one included in the other?

* PolyReg step:

- Random effects are suggested for the baseline and slope, but it is not clear whether this should be used by default or not. Did the authors compare both approaches in terms of performances, stability, computation time, etc.?
- How the significance values are computed and what is the associated null hypothesis? How does it differ from the previous one?
- etc. (most of my questions about the previous steps also applies here)

* Combination step:

- As a whole, the method relies on many procedures, each of them having its own set of parameters. However, as a reader, it is impossible to have a clear picture of the number of parameters, their roles, and the sensitivity of the whole procedure to any of them. This is an issue as a computational method is usable (and in practice correctly used) only when the parameters are clearly described, easy to understand and intuitive enough to tune.

Second, the method is not sufficiently justified:

* RegROTS step:

- Where does the formula to define the default degrees of the polynomials of RegROTS come from? To which extent this value can be considered optimal? How sensitive is the method to the degree defined by the user?
- What is the rationale of the unnumbered equation line 28 page 14? More precisely, why using a geometric mean rather than anything else?
- RegROTS is based on applying ROTS test on permutations resulting from trajectory models. However, to the best of my knowledge (which concurs with the cited references), ROTS has never been applied to this data type. Thus, RegROTS on its own should be evaluated and thoroughly justified.

* DiffROTS step:

- Justification of the geometric means again?

* PolyReg step:

- Why another regression model than in RegROTS is involved, and how is it justified?

- Where does the default degree value formula come from? How does it apply to a wide range of different datasets?

- etc. (most of my questions about the previous steps also applies here)

* Combination step:

- The final methods combine the ranks resulting from the 3 methods, using, once again a geometric mean. What is the rationale for using the geometric mean here rather than anything else? (There is an abundant literature focused on consensus ranking; as well as about combining significance values, and to the best of my knowledge, the geometric mean is not a standard combination procedure).

- To which extent the 3 ranks to be combined are correlated? This is an important question that impacts the FDR procedure (is the FDR control impacted?), but more generally the combination of correlated features in decision-making tends to lead to overfitted results. What are the guarantees against this?

With technical details insufficiently described and a weak theoretical supports, the article gives the impression the reader is invited to blindly accept the new methodology, without understanding its labyrinthine steps, not to speak about questioning it. In my opinion, this contrasts with the classical way of presenting methodological developments in scientific literature.

Third, the method is not fairly evaluated/compared:

- Many classical tools can be used to test a specific trend (H_0) against the absence of this trend, by a likelihood ratio, or an RSS, or possibly, Wald test, and many implementations are already used in biostatistics (e.g., Storey, J. D., Xiao, W., Leek, J. T., Tompkins, R. G. & Davis, R. W. Significance analysis of time course microarray experiments. *Proc. Natl Acad. Sci. USA* 102, 12837–12842, 2005; but many other exists) and a more comprehensive state-of-the-art should be presented and considered in the comparisons.

- The lack of more refined comparisons and evaluation on the various subparts or the methodology makes it impossible to answer the following question: what is the important feature of the methodology? The intensive application of ROTS at any step of the method? Or ROTS is not that important, as the application of any other test on longitudinal models resulting from the proposed regressions would be equally good (which would indicate that the core of the proposed innovation is indeed the extensive use of regressed data)? Or, the combination of multiple ranks resulting from partial approaches (regardless of whether they are based on regression, or not; indicating that for

longitudinal data, one should mainly perform tests in all possible directions, then combine them in a clever way)? Deeper comparisons should be conducted to answer these questions, as their answer would provide a necessary support to the presented methodological developments.

- My last point about fair evaluation is of the utmost importance: From my understanding of the experimental setup, the reference methods were all used on data with N.A.'s, as no imputation algorithm was involved. I understand this is motivated by the fact that the proposed method directly work on raw (i.e, non-imputed) data, so that the reference methods were applied in the same setting. However, this is an issue, as dealing with non-aligned time points or with missing values using regression is essentially an imputation method. In fact, many imputation methods involve regression models between the samples (with or without a temporal link), the peptides or the proteins (within each sample), so that strictly speaking, the propose method embeds an imputation step. It is not named as such, but the same difference. It is not a problem, as it is probably the best way to handle them. However, in this context, stating that the proposed approach is robust to missing values while in fact it only imputes them as in any classically used biostatistics workflow is misleading. And most importantly, comparing the proposed method including its own imputation step with state-of-the-art methods without imputation while they require a separated upstream imputation is clearly incompatible with the standard of scientific publication. I understand this may have happened unnoticed by the authors, but from an experimental viewpoint, this is the same as using a suboptimal tuning or deteriorated versions of reference methods to create an artificial difference of performance: It is called a biased comparison.

Finally, I have a very last concern about the application of the methodology to real-life longitudinal data. Concretely, all the semi-simulated datasets as well as the diabetes dataset display strong temporal patterns. In other words, a time stamp T is strongly correlated to time stamp $T-1$ conditionally to the global trend. While this is not a surprise for the semi-simulated datasets (they were constructed this way), the diabetes dataset cannot be considered a randomly chosen longitudinal analysis. Based on [10], it clearly appears this study was made to highlight temporal profiles. Temporal profiles are of interest, but many longitudinal analyses do not (aim not and cannot) display such highly coherent temporal patterns, for many reasons (fewer time points spread on a longer time range, confounding variables, highly varying patterns, etc.). Of course, having a strong pattern is always more interesting and yields to more trustful conclusions, however many longitudinal analyses are interesting even in absence of such temporal profiles (e.g., long run recovery after a therapy in patient follow-up) for they bring an interesting historical and biological context to interpret the results. Briefly, "longitudinal analyses" are not necessarily "time series", contrarily to those considered in the experimental evaluation. Together with my above concern about the importance of the temporal regression models and their use to impute missing values (or to estimate values at regular time stamps when the data are not synchronized), this becomes alarming. Proteomic experts with few knowledge in temporal/longitudinal data analysis may be tempted to apply the proposed method on a dataset with low temporal coherence and an important rate of missing value. Doing so will exhibit patterns that are more assumed than observed, with the subsequent risk of highlighting false differentially expressed profiles, and thus too optimistic (i.e., erroneous) biological conclusions. As a result, it is essential the authors clearly evaluate, and then

expose the limits of applicability of their methods with respect to the assumption of temporal coherence the method is built on.

For all these reasons, I find this work is premature for publication. As in the current state, I cannot say whether a major revision will be sufficient to exhibit satisfactory results, or on the contrary, if the questions raised above may hinder the interest of the proposed method, I advise the Editor to reject the submission, despite the obvious interest of the subject.

Reviewer: 1

Comments to the Author

This manuscript, titled 'Enhanced longitudinal differential expression detection in proteomics with robust reproducibility optimization regression', proposes that RoIDE, as a robust method applicable to noisy longitudinal proteomics data, combines the detection power of reproducibility optimization and longitudinal regression modeling. The authors clearly show an extensive evaluation of the proposed methods' performance against many existing longitudinal approaches using semi-simulated proteomics datasets with diverse scenarios as well as real experimental biological data, even with non-aligned time points. Also, this proposed method is implemented as an R package in Github and publicly available.

Major issues:

Comment 1:

As the authors address, there are many methods available developed for gene expression microarray data. The authors addressed that these methods do not take into account the characteristics of MS proteomics data. What characteristics of MS proteomics data are not taken into account for these methods? In the introduction, 'missing values' were mentioned as an example. Then, please explain why RoIDE is tolerant for missing values methodologically. For example, RoIDE has a better performance than Timecourse for SGSDS full, which has more missing values. Timecourse performs similarly as RoIDE on other datasets though. The methodological justification for how RoIDE overcome the missing value issue is missing in this version of the manuscript. Also, what other characteristics of MS proteomics data are taken into account for RoIDE

Our response:

We thank the Reviewer for this comment. Indeed, missing values are a common phenomenon in proteomics data discussed in length previously by us and others [17, 18, 49, 50]. Many methods designed for DNA microarrays cannot as such handle missing values at all but require an external imputation method to be used prior to analyzing the data. However, the choice of an imputation method can markedly affect the detection of true differential expression [18]. By using methods able to withstand missing values, the users are relieved from the extra considerations related to the choice and execution of proper imputation in most proteomic datasets with a moderate proportion of missing values.

In this manuscript, we show that the proposed new method RoIDE can stand a moderate number of missing values both in the true positives and true negatives without noticeable effects on the performance. RoIDE performed excellent both in the UPS1 based datasets, where missing values were present only in the true negative background proteins, as well as in the SGSDS based datasets, where missing values were also present in the true positives (spike-in proteins). On the contrary, while the existing method Timecourse also provided very good performance in the UPS1 based datasets, its performance decreased considerably when missing values were present in the true positives in the SGSDS based datasets. Another limitation of Timecourse is that it does not provide significance value estimates for the longitudinal differential expression of the features.

In addition to missing values, another typical characteristic of proteomics data is considerable noise in quantitation [20, 21]. Mass spectrometry data is stochastic by nature and is subject to electrical or chemical

noise due instrumental interference events or contaminants in the soluble sample material. Detecting signal from background noise can be challenging especially for lowly abundant peptides, where the signal is hard to distinguish from the background [20, 21]. In addition, many non-deterministic processes such as miscleavages, ionization competition, ion suppression, and peptide misidentification, can contribute to the formation of missing values and the noisiness of the measurements. Therefore, an effective differential expression method for proteomics data has to be able to withstand considerable noisiness in the measurements in addition to missing values. The same attributes that enable RolDE to perform robustly in the presence of missing values also render it less sensitive to noisy measurements.

RolDE's ability to deal with missing values and noisy measurements can be summarized in the following four features:

- 1) Modularity. RolDE is highly modular, where the composite method is divided into parts. While all of them might not be able to provide a valid score for a protein due to missing values, there might still be enough evidence for detecting possible differential expression.
- 2) Use of ranks. By default, each module assigns a rank for all proteins. If a valid score cannot be determined, such proteins will be placed in random order at the end of the result list, i.e. they will be assigned random ranks worse than those with valid scores. This will naturally place proteins with a large number of missing values at the bottom of the corresponding result list. However, this procedure also allows a protein to achieve better ranks in the other modules where it gets a valid score, thus moving it higher up on the final result list. The rank product approach has been previously shown to be effective for noisy data [47, 48].
- 3) Use of longitudinal regression models. Two of the RolDE modules involve polynomial regression, which can tolerate some number of missing values when estimating the models. Moreover, values for all time points are not necessarily required for individuals to detect evidence for differential expression with enough accuracy.
- 4) Application of the Reproducibility Optimized Test Statistic (ROTS). Two of the RolDE modules utilize ROTS. In addition to noise, it inherently tolerates a moderate number of missing values in the data by effectively ignoring their contribution during the operation of the procedure [28]. In addition to optimizing the reproducibility, ROTS is thus very suitable for analyzing the input data with possible missing values in these modules.

We have now further clarified RolDE's ability to withstand missing values and noise in the measurements in the revised manuscript **page 9 lines 26-45**.

Comment 2:

Please comment about the experiment with less than 5 time points. As the authors mentioned in the introduction, 3 or 4 time points are typical for a biomedical study. The datasets tested in the manuscript have 5, 8, or 9 time points. Can you show the performance for the cases with less than 5 time points or at least comment it?

Our response:

We thank the Reviewer for pointing this out. We have now further compared the performance of RolDE together with the best-performing existing methods Timecourse and Limma and the baseline ROTS method in 1200 semi-simulated datasets with three or four time points. With both numbers of time points, we generated 300 semi-simulated UPS1 based filtered datasets with no missing values and 300 semi-simulated UPS1 full datasets including missing values to comprehensively evaluate the performance of the methods in longitudinal datasets with only few time points. Consistent with the results in datasets with more timepoints, RolDE performed overall best in all scenarios, while the established longitudinal method Timecourse also performed well and outperformed the baseline method ROTS. These comparisons are now added to the revised manuscript (new **Supplementary Figure 2, page 6 lines 3-13**).

Minor issues:

Comment 3:

The supplementary figure numbers seem not matched in the main text. Need to double-check them : For example, 1) page 7, line 44 : suppl fig 2 ->3, 2) Page 15, line 33 : suppl fig 4A -> 1A 3) Page 16, line 5 : suppl fig 4B -> 1B?

Our response:

We thank the Reviewer for noticing these errors. We have **corrected these** for the revised manuscript.

Comment 4:

Please clarify what page 10, line 27, 'what kinds of differences are searched from the data' means.

Our response:

We apologize for the ambiguous wording. The sentence means that the user does not need to provide RolDE any prior information on the nature of what type of differential expression patterns (e.g. stable, linear, quadratic, cubic, sigmoid, etc.) are searched from the data. We have now clarified this in the revised manuscript (**page 11, lines 20-23**).

Comment 5:

Suggest submitting RolDE R package in GitHub to Bioconductor. It manages all dependencies and reviews the codes for the R package, which supports the long-lifetime for R packages and usability.

Our response:

We thank the Reviewer for this suggestion. We agree that Bioconductor promotes the usability and accessibility of R packages related to the high-throughput biological data analysis field. We have now submitted the RolDE R package to Bioconductor.

Reviewer: 2

Comments to the Author

The manuscript describes a statistical methodology to find differentially expressed proteins in longitudinal proteomics analysis. It is an important and timely subject, for which new results are needed in the proteomics community.

The proposed approach is based on interesting rationale and show interesting preliminary results on benchmarks datasets. However, in its current state, the manuscript failed to convince me. As is, the article is focused on the results obtained on the benchmark datasets, as if they were discoveries of interest, while their only purpose is to support a new method, which (too) short presentation is postponed to the method section (together with the MS routines and state of the art methods), and which exhaustive evaluation is missing. Hereafter follows a more detailed description of my concerns about (i) the insufficient description; (ii) the insufficient justification; (iii) the biased comparative evaluation; (iv) the scope of application of the proposed methodology.

Comment 1:

What family of orthogonal polynomials are used in RegROTS?

Our response:

The orthogonal polynomials in the RegROTS module are generated using the three term recursion algorithm described in [67, pp.343-4]. This is the default way of generating orthogonal polynomials in R using the function `poly`. If the time points in the compared conditions are aligned, we use the default algorithm. If the time points are not aligned, we omit the last scaling step to preserve comparability between the coefficients of the individual regression models, while the polynomials remain orthogonal. This is now clarified in the revised manuscript (**page 15, lines 31-41**).

Comment 2:

How does one switches from the $\Delta\beta$'s to the R_i 's of the unnumbered equation line 28 page 14?.

Our response:

We thank the Reviewer for this question. The individuals in the different conditions are first divided into runs for statistical testing so that each individual is used only once in each run. Within each run, multigroup ROTS is used to test the null hypothesis $\Delta\beta_0 = \Delta\beta_1 = \dots = \Delta\beta_d = 0$ for each protein to investigate, whether the differences in the regression coefficients of the individual models between the different conditions are effectively zero, and the proteins are then ranked according to the significance values of the modified F-statistic from the multigroup ROTS. The final score for each protein, shown in the equation, is calculated by combining these ranks from the different runs using the rank product (i.e. the geometric mean). We have now further clarified this in the revised manuscript (**page 15, lines 42-44 and page 16, lines 1-20**). We have also numbered the equations for easier reference.

Comment 3:

In the text above the equation, it is indicated that the significance value of the R_i 's are used, while the equation directly refers to the R_i 's, not their significance values. This needs clarification.

Our response:

Indeed, as the Reviewer correctly points out, the equation indicates that the ranks r_i from the different RegROTS runs i are used for calculating the final score, not the significance values as such. The ranks are determined based on the significance values of the modified F -statistic from the multigroup ROTS in each run, with the most significant protein assigned rank 1. We have now clarified this in the revised manuscript (**page 16, lines 10-15**).

Comment 4:

More generally, the entire paragraph above this equation only provides a hand-waving description, or an example that is not sufficient to thoroughly understand the methodology or to re-implement it.

Our response:

We apologize for the confusion. We have now further clarified the description of the RegROTS module in the revised manuscript to clarify the methodology (**pages 15-16**). To ensure reproducibility, we also provide all the codes in the publicly available R package, allowing detailed investigation of our current implementation.

Comment 5:

What is the empirical distribution of S_{RegROTS} under the null hypothesis?

Our response:

We have now included the empirical distribution of S_{RegROTS} in new **Supplementary Figure 8A**, described in the revised manuscript (**page 16, lines 17-18**).

Comment 6:

What is the concrete meaning of the proposed null hypothesis? (i.e., the equality of the $\Delta\beta$'s to zero)?

Our response:

The null hypothesis $\Delta\beta_0 = \Delta\beta_1 = \dots = \Delta\beta_d = 0$ means that there are no differences in any of the regression coefficients of the individual models between the different conditions. More specifically, there are no differences related to the intercept $\Delta\beta_0$ (mean expression), linear longitudinal trends $= \Delta\beta_1$ (slope), quadratic trends $= \Delta\beta_2$, or any of the examined polynomial trends until degree d ($\Delta\beta_d$). This indicates that the regression models for the individuals in the different conditions are very similar to each other. We have now clarified this in the revised manuscript (**page 16, lines 18-20**).

Comment 7:

DiffROTS step:

- How missing values are dealt with? It looks like they are simply considered similarly as unaligned values, so that they are replaced by regression estimates... This must be clarified.

Our response:

We thank the Reviewer for pointing this out. The DiffROTS module directly examines the expression differences between the conditions at each time point. Thus, no regression-based estimates are used in the DiffROTS module in case of aligned time points. If any of the protein abundance values are missing, this simply results in a missing value for that time point in the analysis. If the time points in the data are not aligned between the individuals and/or the conditions, the expression levels at different conditions cannot be directly compared at different time points. In that case, two group ROTS is used instead of multigroup ROTS to explore longitudinal differential expression between the conditions after accounting for longitudinal trends of varying complexity in the data with polynomial regression. However, even in the case of non-aligned time points, missing values remain as missing values in the DiffROTS module, since missing expression values will result in missing residual values after accounting for the longitudinal trends. We have now further clarified this in the revised manuscript (**page 16, lines 26-27, page 17, lines 1-16 and page 9, lines 36-38**).

Comment 8:

What is the polynomial regression at use? (RegROTS or PolyReg one) why?

Our response:

By default, no regression is used in the DiffROTS module, but the expression levels are compared directly. However, if the time points are not aligned, polynomial regression is used to account for longitudinal trends of varying complexity in the data. We have now clarified this and further expanded the description of the DiffROTS module in the case on non-aligned time points in the revised manuscript (**page 17, lines 1-16**).

Comment 9:

Same ambiguity as with RegRots about R_i 's or their significance values.

Our response:

We apologize for the confusion. Similarly as with the RegROTS module, the ranks r_i from the different DiffROTS runs i are used for calculating the final score for each protein. The ranks are determined based on the significance values of the modified F -statistic from the multigroup ROTS in each run, with the most significant protein assigned rank 1. Please also see our response to Comment 3. Unlike with RegROTS, however, here the exception are datasets with non-aligned time points, where the DiffROTS module is adjusted to such a design and the significance values are used for determining the score. We have now further clarified this in the revised manuscript (**page 16, lines 31-37 and page 17, lines 1-16**).

Comment 10:

What is the empirical distribution of $S_{\{\text{DiffROTS}\}}$ under the null hypothesis?

Our response:

We now included the empirical distribution of $S_{DiffROTS}$ in the new **Supplementary Figure 8B**, described in the revised manuscript (**page 16, lines 37-38**).

Comment 11:

To which extent the null hypothesis from DiffROTS differs from that of RegROTS? Is one included in the other?

Our response:

The null hypothesis of RegROTS is $\Delta\beta_0 = \Delta\beta_1 = \dots = \Delta\beta_d = 0$, that is, there are no differences in the coefficients of the individual regression models between the different conditions (see also our response to Comment 6). Instead, the null hypothesis of DiffROTS is $\Delta y_1 = \Delta y_2 = \dots = \Delta y_T = 0$, that is, there are no differences in the expression values at the distinct time points between the different conditions. Thus, the null hypotheses of the different modules are different. While both are related to differential expression between the conditions, they complement each other well in practice, as there is always uncertainty and noise in determining whether the null hypothesis is rejected or accepted.

Comment 12:

PolyReg step:

- Random effects are suggested for the baseline and slope, but it is not clear whether this should be used by default or not. Did the authors compare both approaches in terms of performances, stability, computation time, etc.?

Our response:

We thank the Reviewer for pointing this out. By default, RolDE uses fixed effects models for the PolyReg module when the time points are aligned and mixed effects models with random effects for the individual baseline when the time points are not aligned. However, the user can easily control the used models in the PolyReg module by changing the parameter `model_type` in RolDE and selecting the appropriate random effects (intercept and slope). The application of mixed effects models instead of fixed only models increases the required computational time depending on the analyzed dataset (e.g. 5-30 seconds in the UPS1 datasets with 1581 proteins and 30 samples). In principle, it also requires more data to account for the individual variability in the mixed model approach and can thus be more sensitive to missing values. In the tested semi-simulated spike-in datasets, however, no major differences were observed in the performance of RolDE when using only fixed effect models or mixed effects models with a random effect for the individual baseline in the PolyReg module (new **Supplementary Figure 3**). Mixed effects models should be considered especially when heterogeneity and uneven sampling points (non-aligned time points) are expected among individuals [68], which is the case in many real-life studies. We have further clarified this in the revised manuscript (**page 17, lines 23-28**).

Comment 13:

PolyReg step:

How the significance values are computed and what is the associated null hypothesis? How does it differ from the previous one? - etc. (most of my questions about the previous steps also applies here)

Our response:

In the PolyReg module, for each protein, a polynomial regression model is fitted, involving both time and condition related variables. For each of the condition related coefficients γ_j , the null hypothesis is $\gamma_j = 0$, that is, the condition related variable is not statistically significantly associated with the abundance of the protein. Each null hypothesis is examined using *t*-test, which is the standard way of assessing the significance of regression coefficients. The corresponding significance values p_{γ_j} are then used to determine the final score for the PolyReg module, $S_{PolyReg}$, which is calculated as the minimum over the significance values of the condition related regression coefficients. In the PolyReg module, the polynomial regression models are constructed using all replicates across all conditions in a single model, which complements the RegROTS and DiffROTS modules focusing on individual regression models in the different conditions and condition-specific expression differences at distinct time points, respectively (see also our responses to Comments 6 and 11). Thus, the complementary null hypotheses of the different modules support well robust evaluation of longitudinal differential expression in practice, as illustrated by our extensive benchmarking. We have now clarified this and the details of how the PolyReg module calculates the significance values in the revised manuscript (**page 17, lines 29-36**).

Comment 14:

Combination step:

- As a whole, the method relies on many procedures, each of them having its own set of parameters. However, as a reader, it is impossible to have a clear picture of the number of parameters, their roles, and the sensitivity of the whole procedure to any of them. This is an issue as a computational method is usable (and in practice correctly used) only when the parameters are clearly described, easy to understand and intuitive enough to tune.

Our response:

We fully agree with the Reviewer that usability of a computational method is important, including the use of different parameters. While, indeed, RolDE combines the power of its three modules when detecting longitudinal differential expression, the default usage of RolDE is very easy. Although the user is allowed full control of the parameters through an explicit function, at bare minimum, the user needs to provide only the data, the design matrix, and information on whether the time points in the data are aligned (TRUE by default). The design matrix needs to contain the sample names, the condition information for each sample, the time information for each sample, and the information from which individual the sample is from. All the comparisons in the manuscript were run with these default settings without any further tuning of the method. As can be seen from our results, RolDE performed excellent with these default parameter settings across all the different datasets.

The optional parameters of RolDE that the user can modify include 1) use of random effects for the individual baseline or slope in the PolyReg module, 2) the polynomial degree in the RegROTS and PolyReg modules. No major differences were observed in the performance of RolDE when using fixed effect models or mixed effects models with a random effect for the individual baseline in the PolyReg module (see also our response to Comment 12). We have also tested the sensitivity of RolDE to the polynomial degrees in

the RegROTS and PolyReg modules, suggesting that RolDE is not sensitive to the specific degrees used in these modules. To demonstrate this, we have now included comparisons of RolDE with various degrees for the RegROTS and PolyReg modules in the revised manuscript (new **Supplementary Figure 3**). Thus, overall, the composite method RolDE is a robust approach and is not sensitive to slight variations in its parameters. The default parameters of RolDE offer a good overall performance in a wide variety of different datasets, while the user can (but is not required to) control the details of the procedure. The investigation of the robustness of RolDE is now included in the revised manuscript (**page 6, lines 14-47, page 11, lines 14-17**).

Comment 15:

Second, the method is not sufficiently justified:

* RegROTS step:

- Where does the formula to define the default degrees of the polynomials of RegROTS come from? To which extent this value can be considered optimal? How sensitive is the method to the degree defined by the user?

Our response:

In our automatic approach, the degrees for the RegROTS module were selected to vary between 1 and 4, depending on the number of time points in the data. The degrees for the PolyReg module were selected to vary between 2 and 5, depending on the number of time points in the data, while being also higher than those in the RegROTS module. The maximum degrees for the RegROTS and the PolyReg modules were limited to 4 and 5, respectively, to prevent overfitting and also due to the fact that, in practice, such polynomial degrees are typically sufficient to detect all kinds of polynomial trend differences even in data with many time points.

We have tested the sensitivity of RolDE to the polynomial degrees in both the RegROTS and PolyReg modules, suggesting that RolDE is not sensitive to the specific degrees used in these modules. In practice, our results suggest that using different polynomial degrees for the RegROTS and PolyReg modules works best, which also supports the complementarity of the modules; the RegROTS module focuses mainly on trends of slightly lower complexity, while the PolyReg module considers a broader range of trend differences. While the performance of RolDE was not sensitive to the choice of the degrees of the polynomials, our default automatic approach was among the top performers when comparing different degree combinations for the RegROTS and PolyReg modules in the nearly 2000 semi-simulated spike-in datasets (new **Supplementary Figure 3, page 6, lines 15-28, and page 11, lines 14-17**).

Comment 16:

What is the rationale of the unnumbered equation line 28 page 14? More precisely, why using a geometric mean rather than anything else?

Our response:

RolDE relies on the use of ranks and rank products (i.e., the geometric mean of ranks) to combine the different runs and modules. The rank product method is motivated by the common biological assumption that if a feature is repeatedly at the top of the lists in multiple comparisons, it is more likely to be a true finding. In the context of omics studies, the rank product method was initially developed for DNA

microarrays for the detection of differentially expressed genes [Breitling R, Armengaud P, Amtmann A, Herzyk P (2004) Rank products: a simple, yet powerful, new method to detect differentially regulated genes in replicated microarray experiments. *FEBS Lett* 573(1):83–92]. Since then, the rank product has become a popular choice for the analysis of various omics datasets, including proteomics [2, Smit S, van Breemen MJ, Hoefsloot HCJ, Smilde AK, Aerts JMFG, de Koster CG (2007) Assessing the statistical validity of proteomics based biomarkers. *Anal Chim Acta* 592(2):210–217; Schwämmle V, León IR, Jensen ON (2013) Assessment and improvement of statistical tools for comparative proteomics analysis of sparse data sets with few experimental replicates. *J Proteome Res* 12(9):3874–3883; Koziol JA (2010) The rank product method with two samples. *FEBS Lett* 584(21):4481–4484]. The rank product statistic has been noted to be particularly useful for the analysis of noisy datasets and for small numbers of replicates, as it does not rely on any distributional assumptions of the data [47, 48, Breitling R, Armengaud P, Amtmann A, Herzyk P (2004) Rank products: a simple, yet powerful, new method to detect differentially regulated genes in replicated microarray experiments. *FEBS Lett* 573(1):83–92; Heskes T, Eisinga R, Breitling R (2014) A fast algorithm for determining bounds and accurate approximate *p*-values of the rank product statistic for replicate experiments. *BMC Bioinformatics* 15(1):367]. The motivation to use the rank product (geometric mean) in RolDE thus has its origins in these previous research work. We have elucidated this in the revised manuscript on **page 9 lines 32-35** and **page 16 lines 15-17**. To further clarify the usage of terms, we have now changed the geometric mean to rank product throughout the revised manuscript, which is perhaps a more familiar term instead of geometric mean.

Comment 17:

RegROTS is based on applying ROTS test on permutations resulting from trajectory models. However, to the best of my knowledge (which concurs with the cited references), ROTS has never been applied to this data type. Thus, RegROTS on its own should be evaluated and thoroughly justified.

Our response:

We thank the Reviewer for this comment. Indeed, ROTS has typically been applied in the context of two-group expression data for different omics types [12, 26, 27, 28, 29]. However, ROTS is a data adaptive method which can optimize its parameters based on intrinsic features of input data. The input data for ROTS within the RegROTS module consists of $\Delta\beta$ values from the individual regression models (see also our responses to Comments 2 and 6). We have now further evaluated the performance of RegROTS separately on all the 1920 semi-simulated datasets (new **Supplementary Figure 4**). Furthermore, we have compared the performance of RegROTS to the same procedure where ROTS has been replaced with the traditional one-way-analysis-of-variance (ANOVA) approach (new **Supplementary Figure 4**). The results clearly demonstrate the benefits of applying ROTS over the more traditional ANOVA approach (Wilcoxon signed-rank test $p < 10^{-6}$ in all scenarios). These new results are described in the revised manuscript on **page 6 lines 29-36**.

Comment 18:

DiffROTS step:

- Justification of the geometric means again?

Our response:

Please see our response to Comment 16.

Comment 19:

PolyReg step:

- Why another regression model than in RegROTS is involved, and how is it justified?

Our response:

In the PolyReg module, for each protein, a polynomial regression model is fitted, involving both time and condition related variables, using all individuals across all conditions in a single model. This complements the RegROTS and DiffROTS modules focusing on individual regression models in the different conditions and condition-specific expression differences at distinct time points, respectively (see also our responses to Comments 6, 11, and 13). Although both the RegROTS and the PolyReg modules apply polynomial regression models, they are used very differently in detecting longitudinal differential expression. For instance, while the RegROTS module might be more sensitive to individual differences in longitudinal expression, it can also be more sensitive to individual noise in the data. Thus, the different modules support robust evaluation of longitudinal differential expression in practice, as illustrated by our extensive benchmarking. This is particularly important in the presence of missing values and noise, which are typical in many proteomics studies. We have now clarified this in the revised manuscript (**page 6, lines 37-39 and page 9, lines 26-32, new Supplementary Figure 4**).

Comment 20:

PolyReg step:

Where does the default degree value formula come from? How does it apply to a wide range of different datasets?

Our response:

Please see our response to Comment 15.

Comment 21:

* Combination step:

- The final methods combine the ranks resulting from the 3 methods, using, once again a geometric mean. What is the rationale for using the geometric mean here rather than anything else? (There is an abundant literature focused on consensus ranking; as well as about combining significance values, and to the best of my knowledge, the geometric mean is not a standard combination procedure).

Our response:

RoLDE relies on the use of ranks and rank products (i.e., the geometric mean of ranks) to combine the different modules. The rank product method is motivated by the common biological assumption that if a feature is repeatedly at the top of the lists in multiple comparisons, it is more likely to be a true finding. In the context of omics studies, the rank product method was initially developed for DNA microarrays for the detection of differentially expressed genes [Breitling R, Armengaud P, Amtmann A, Herzyk P (2004) Rank

products: a simple, yet powerful, new method to detect differentially regulated genes in replicated microarray experiments. FEBS Lett 573(1):83–92]. Since then, the rank product has become a popular choice for the analysis of various omics datasets, including proteomics [2, Smit S, van Breemen MJ, Hoefsloot HCJ, Smilde AK, Aerts JMFG, de Koster CG (2007) *Assessing the statistical validity of proteomics based biomarkers. Anal Chim Acta 592(2):210–217*; Wiederhold E, Gandhi T, Permentier HP, Breitling R, Poolman B, Slotboom DJ (2009) *The yeast vacuolar membrane proteome. Mol Cell Proteomics 8(2):380–392*; Schwämmle V, León IR, Jensen ON (2013) *Assessment and improvement of statistical tools for comparative proteomics analysis of sparse data sets with few experimental replicates. J Proteome Res 12(9):3874–3883*; Koziol JA (2010) *The rank product method with two samples. FEBS Lett 584(21):4481–4484*] as well as meta-analysis [Hong F, Breitling R, McEntee CW, Wittner BS, Nemhauser JL, Chory J (2006) *RankProd: a bioconductor package for detecting differentially expressed genes in meta-analysis. Bioinformatics 22(22):2825–2827*]. The rank product statistic has been noted to be particularly useful for the analysis of noisy datasets [47, 48]. The motivation to use the rank product (geometric mean) in RolDE thus has its origins in these previous research work. We have elucidated this in the revised manuscript (on **page 9 lines 32-35 and page 16 lines 15-17**). To further clarify the usage of terms, we have now changed the geometric mean to rank product throughout the revised manuscript, which is perhaps a more familiar term instead of geometric mean.

Comment 22:

To which extent the 3 ranks to be combined are correlated? This is an important question that impacts the FDR procedure (is the FDR control impacted?), but more generally the combination of correlated features in decision-making tends to lead to overfitted results. What are the guarantees against this?

Our response:

We thank the Reviewer for the comment. Indeed, to some extent the ranks from the different modules will be correlated in a given data and part of the consistently good performance and robustness of RolDE is attributable to the use of correlated ranks. True positives tend to receive good ranks in each module, while false positives are more module-specific due to the diverse approaches used. Then, combining the modules via the rank product naturally results in propagation of small rank products for the true positives, while the more module-specific false positives receive larger overall rank product and trickle downwards in the result list of the composite method.

To take the correlation of the rankings into account when estimating the statistical significance of the findings, the significance values for RolDE are estimated through a simulation procedure. By taking into account the dependencies between the runs in the RegROTS and DiffROTS modules and the different polynomial coefficients in the PolyReg module, the simulation procedure results in simulated ranks that are similarly correlated as in the evaluated experimental data. Under the null hypothesis, the simulated rank products correspond well to the experimental rank products of RolDE (**Supplementary Figure 8C**) and the estimated significance values are uniformly distributed, as expected (**Supplementary Figure 8D**). After the adjustment for multiple hypothesis testing, significant findings under the null hypothesis at false discovery rate (FDR) of 0.05 (using the Benjamini-Hochberg method [69]) are rare and within the limits of the defined FDR rate (**Supplementary Table 1**). We have now further clarified the estimation of the significance values and how the dependencies between the different modules are taken into account in the revised manuscript (**page 18, lines 2-32**). We have also included our extensive demonstration of the effectiveness of RolDE in controlling the number of false positives using altogether 600 datasets of various types with and

without different proportions of missing values under the null hypothesis (new **Supplementary Table 1, page 7, lines 4-11 and page 13, lines 29-44**). The results clearly demonstrated that, in the absence of a true longitudinal differential expression signal in the data, RolDE will not generate large amounts of erroneous false positive findings.

Overall, the performance of RolDE, together with 15 other methods, was evaluated in 1920 semi-simulated spike-in datasets with various types of longitudinal trends and trend differences using ROC analysis, considering the sensitivity (i.e. true positive rate) against the specificity (i.e. true negative rate). The area under the ROC curve (AUC) was used to measure how well each method was able to distinguish the true signal of interest when the detection threshold (e.g. significance value) was varied. If significant overfitting by the method for a given dataset would occur, the method would not perform consistently well in this evaluation, where diverse datasets are explored and the true positives, false positives, true negatives, and false negatives at different detection thresholds affect the ROC curve and AUC. Thus, our extensive performance evaluations in the 1920 semi-simulated spike-in datasets demonstrate RolDE's ability to correctly detect true longitudinal differential expression in the presence of various types of longitudinal trends, trend differences and dataset types. Importantly, the additional 600 datasets of various types with and without different proportions of missing values under the null hypothesis further show that when a true longitudinal differential expression signal is not present in the dataset, RolDE will not generate artificial findings (false positives).

Comment 23:

With technical details insufficiently described and a weak theoretical supports, the article gives the impression the reader is invited to blindly accept the new methodology, without understanding its labyrinthine steps, not to speak about questioning it. In my opinion, this contrasts with the classical way of presenting methodological developments in scientific literature.

Our response:

We apologize for this confusion. We have now further clarified the proposed method RolDE and the used methodology in the revised manuscript (**page 15, lines 31-44, page 16, lines 1-20, page 16, lines 26-40, page 17, lines 1-16, page 17, lines 23-36 and page 18, lines 13-22**).

Comment 24:

Third, the method is not fairly evaluated/compared:

- Many classical tools can be used to test a specific trend (H0) against the absence of this trend, by a likelihood ratio, or an RSS, or possibly, Wald test, and many implementations are already used in biostatistics (e.g., Storey, J. D., Xiao, W., Leek, J. T., Tompkins, R. G. & Davis, R. W. Significance analysis of time course microarray experiments. Proc. Natl Acad. Sci. USA 102, 12837–12842,2005; but many other exists) and a more comprehensive state-of-the-art should be presented and considered in the comparisons.

Our response:

We thank the Reviewer for this suggestion. To make our comparisons even more comprehensive, we have included additional methods in the comparisons for the revised manuscript as suggested by the Reviewer. We have included the *Extraction of Differential Gene Expression (EDGE)*, suggested by the reviewer [73], as

well as more state of the art methods *A Linear Mixed Model Spline Framework for Analysing Time Course 'Omics' Data (LMMS)* and [74] *Omics Longitudinal Differential Analysis (OmicsLonDA)* [75]. While all the additional methods displayed good performance in the evaluated datasets, our results demonstrated the superior overall performance of RolDE across the datasets. The new results have been added throughout the revised manuscript.

Comment 25:

The lack of more refined comparisons and evaluation on the various subparts or the methodology makes it impossible to answer the following question: what is the important feature of the methodology? The intensive application of ROTS at any step of the method? Or ROTS is not that important, as the application of any other test on longitudinal models resulting from the proposed regressions would be equally good (which would indicate that the core of the proposed innovation is indeed the extensive use of regressed data)? Or, the combination of multiple ranks resulting from partial approaches (regardless of whether they are based on regression, or not; indicating that for longitudinal data, one should mainly perform tests in all possible directions, then combine them in a clever way)? Deeper comparisons should be conducted to answer these questions, as their answer would provide a necessary support to the presented methodological developments.

Our response:

We agree with the Reviewer that showing more refined comparisons would provide deeper understanding about the benefits of the RolDE approach. To demonstrate the benefits of ROTS, we have now composed and compared a variant of the RegROTS module utilizing a standard one-way Analysis of Variance (ANOVA) instead of ROTS. We have extensively compared this RegANOVA method to the RegROTS module in all the 1920 semi-simulated datasets. The comparisons clearly demonstrated the benefits of applying ROTS over ANOVA with considerable performance gains (Wilcoxon signed-rank test $p < 10^{-6}$ in all scenarios). These results are included in the new **Supplementary Figure 4** of the revised manuscript and described on **page 6, lines 29-36**.

While the RegROTS module alone performed very well, the use of complementary approaches in RolDE increased the performance further. Ultimately, the combination of the diverse approaches through ranks and rank products stabilizes the composite method and allows for consistent excellent performance in diverse datasets (**Supplementary Figure 4, page 6, lines 37-41**).

While combining any multiple diverse approaches might be a valid strategy, as suggested by the Reviewer, it does not always result in improvements in the performance, but can also reduce the performance. To demonstrate this, we have now combined three diverse methods Timecourse, Limma and MaSigPro using a similar ranking and rank product approach as in RolDE in two different types of datasets, the semi-simulated SGSDS full and the UPS1 mix full datasets (new **Supplementary Figure 5, page 6, lines 41-47 and page 7, lines 1-3**). Indeed, the composite approach of Timecourse, Limma and MaSigPro performed relatively well. However, in the UPS1 mix datasets, the performance was significantly reduced, when compared to Timecourse alone (Wilcoxon signed-rank test $p < 10^{-15}$).

Comment 26:

My last point about fair evaluation is of the utmost importance: From my understanding of the experimental setup, the reference methods were all used on data with N.A.'s, as no imputation algorithm

was involved. I understand this is motivated by the fact that the proposed method directly work on raw (i.e, non-imputed) data, so that the reference methods were applied in the same setting. However, this is an issue, as dealing with non-aligned time points or with missing values **using regression is essentially an imputation method**. In fact, many imputation methods involve regression models between the samples (with or without a temporal link), the peptides or the proteins (within each sample), so that strictly speaking, the propose method embeds an imputation step. It is not named as such, but the same difference. It is not a problem, as it is probably the best way to handle them. However, in this context, stating that the proposed approach is robust to missing values while in fact it only imputes them as in any classically used biostatistics workflow is misleading. And most importantly, comparing the proposed method including its own imputation step with state-of-the-art methods without imputation while they require a separated upstream imputation is clearly incompatible with the standard of scientific publication. I understand this may have happened unnoticed by the authors, but from an experimental viewpoint, this is the same as using a suboptimal tuning or deteriorated versions of reference methods to create an artificial difference of performance: It is called a biased comparison.

Our response:

We agree with the Reviewer that fair evaluation is of utmost importance. Therefore, we included in the evaluation a large number of both *full* and *filtered* datasets of different types. In the *filtered* datasets, all proteins with missing values were filtered out before analysis so that the *filtered* datasets contained no missing values whatsoever. Thus, all the methods were evaluated on altogether 810 datasets without any missing values. These comparisons were not affected in any way by missing values nor the lack of imputation thereof.

In addition, all the methods that tolerated missing values were analyzed in the same 810 datasets without any filtering and an additional 300 CPTAC based datasets with missing values to investigate how the methods performed in the presence of missing values (the *full* datasets). In this case, the methods were evaluated without imputation, as we and others have shown the importance of careful consideration regarding imputation [18]. We agree with the Reviewer that regression can also be considered essentially an imputation method and apologize for the confusion. However, we did not consider the regression used in the RolDE modules as an imputation approach as such and, technically, no imputation of missing values *per se* was done as opposed to actual imputation approaches (regression-based or other), where the values are directly imputed. Rather the regression models can be estimated even when missing values are present. In fact, the use of regression models in some form is also true for most of the methods evaluated in this study. Of the methods used in the *full* datasets with missing values, only Timecourse is not based on regression. Thus, most of the methods should be able to deal with missing values in the data in a similar fashion to RolDE.

We have now further clarified the two different types of datasets tested without and with missing values throughout the revised manuscript. The use of regression in the presence of missing values is clarified on **page 9, lines 35-41**.

Comment 27:

Finally, I have a very last concern about the application of the methodology to real-life longitudinal data. Concretely, all the semi-simulated datasets as well as the diabetes dataset display strong temporal patterns. In other words, a time stamp T is strongly correlated to time stamp T-1 conditionally to the global

trend. While this is not a surprise for the semi-simulated datasets (they were constructed this way), the diabetes dataset cannot be considered a randomly chosen longitudinal analysis. Based on [10], it clearly appears this study was made to highlight temporal profiles. Temporal profiles are of interest, but many longitudinal analyses do not (aim not and cannot) display such highly coherent temporal patterns, for many reasons (fewer time points spread on a longer time range, confounding variables, highly varying patterns, etc.). Of course, having a strong pattern is always more interesting and yields to more trustful conclusions, however many longitudinal analyses are interesting even in absence of such temporal profiles (e.g., long run recovery after a therapy in patient follow-up) for they bring an interesting historical and biological context to interpret the results. Briefly, "longitudinal analyses" are not necessarily "time series", contrarily to those considered in the experimental evaluation. Together with my above concern about the importance of the temporal regression models and their use to impute missing values (or to estimate values at regular time stamps when the data are not synchronized), this becomes alarming. Proteomic experts with few knowledge in temporal/longitudinal data analysis may be tempted to apply the proposed method on a dataset with low temporal coherence and an important rate of missing value. Doing so will exhibit patterns that are more assumed than observed, with the subsequent risk of highlighting false differentially expressed profiles, and thus too optimistic (i.e., erroneous) biological conclusions. As a result, it is essential the authors clearly evaluate, and then expose the limits of applicability of their methods with respect to the assumption of temporal coherence the method is built on.

Our response:

We thank the Reviewer for raising this important issue. The longitudinal type 1 diabetes (T1D) proteomics data of [10] included 2037 proteins after preprocessing. Among these proteins, the majority of the proteins did not show any temporal patterns, as stated also in the original publication by Liu et al. [10]: "majority of the protein groups have the temporal expression trend of flat i.e. age-independent expressions". Only two proteins with differential expression between T1D cases and controls at FDR of 0.05 were discovered in the original study, presented in their Supplementary Table 3 [10]. Thus, the explored longitudinal T1D proteomics data for most part did not display strong temporal patterns but consisted mostly of highly varying, highly noisy longitudinal measurements of serum protein expression levels over time, typically spanning a period of 12-13 years without fixed time intervals. In general, T1D omics data and especially serum proteomics data are considered to be difficult data to analyze due to high variability of the measurements, possible underlying confounding variables, the complexity of the disease pathogenesis, the large dynamic range of circulating proteins, and the diversity of known and unknown protein isoforms [Ignjatovic V, Geyer PE, Palaniappan KK, et al (2019) *Mass Spectrometry-Based Plasma Proteomics: Considerations from Sample Collection to Achieving Translational Data. J Proteome Res* 18(12):4085–4097]. Thus, the T1D data is an example of a complex real-life follow-up study using proteomics.

Regarding missing values or unsynchronized time points, we apologize for the confusion and unclear description. The regression approaches in RolDE are not used to impute any specific measurement values as such but they are only used to evaluate differences in the overall expression levels and trends in the compared conditions. For the RegROTS and PolyReg modules, this is true as such, even if the time points are not synchronized, whereas the DiffROTS module is adjusted to such a design using polynomial regression to adjust for overall time trends. Again, no values are imputed. We have now substantially clarified this in the revised manuscript (**page 17, lines 1-16**).

Finally, we very much agree with the Reviewer that it is important to avoid erroneous conclusions. Therefore, we have now extensively assessed the effectiveness of RolDE in controlling false discoveries (see also our response to Comment 23). Our results demonstrate that, in the absence of true differential expression patterns, false detections are rare. However, it should be noted that RolDE must be applied for properly normalized and properly adjusted data. RolDE does not perform any normalization or adjustment for the data for any possible confounding effects. If such are present in the data, they must be removed prior to the application of RolDE. Furthermore, RolDE is intended specifically for detecting longitudinal differential expression between two conditions in high-throughput data and is not suitable for other types of longitudinal analysis (e.g. looking for overall trends over both conditions, searching for specific kind of trends/changes in the data, analyzing longitudinal clinical or other than high-throughput data). RolDE is intended as a general discovery tool for longitudinal differential expression patterns from high-throughput data, especially messy proteomics data. As with all such tools and in typical omics studies, the top findings should be carefully considered, visualized and validated via other more targeted approaches (such targeted MS, Western Blot, etc.). We have further discussed these and other possible limitations of RolDE in the revised manuscript, **page 11, lines 26-31**.

Reviewers' comments:

Reviewer #2 (Remarks to the Author):

To complete this revised version, the authors performed an impressive amount of experimental work and tentatively answer most of my remarks from the first round. I acknowledge them for doing so, yet, the revised version failed to convince me in broadly the same terms as the first one. Although few of the authors' answers were highlighting, some raised new concerns and most confirmed my general opinion after reading the first version.

This now confirmed opinion is the following: the article describes (only partially and in vague literate terms in place of algorithms and formulae) a rather labyrinthine procedure, which various elements are weakly supported (either by theoretical results or by well-formulated rationale; as opposed to extensive empirical evaluations on simulated data). It does not mean that it cannot deal with the task it was originally designed for, and in fact, it is probably the case. However, as smart as it is, it is not sufficient to be considered a methodological development in biostatistics. This is the main difference between the engineering tricks and workflows that sometimes accompanies the M&M section of an article describing new biological discoveries, and a methodological development that is worth of publication as a self-contained novelty. Notably and as already explicated in my former review, in the latter case, the article must provide an exhaustive and accurate description/justification/evaluation of every single step. The authors largely extended the volume of their simulations to address the last point, however: (1) it remains difficult to assess the extent to which these simulations fit with their algorithmic implicit assumptions (so that it is legitimate to wonder on the generalization capabilities on different real data), and (2) at some points repeated simulations cannot cope for the lack of theoretical guarantees. As for the two other points (description and justification), the article changes are not sufficient, notably:

- the list of parameters (those accessible to the users as well as those which can be tuned to a default value) still fails to appear in a synthetic way in the article; and the entire necessary information about their precise influence, their individual stability, their range of acceptable values, etc. is only partially provided by extensive yet too global comparisons for which no clear material can be extracted.

- the very details of the algorithm (possibly through the form of an algorithm or a detailed flowchart) is still missing. Worse, I discovered when reading the rebuttal that depending on the timepoint being align or not, or depending on the presence of MV, some subroutines in various modules may change their processing, in a way that is not described in the article. To me, this more a source of concern than anything else. Worst, in their answer, the authors seems to assume that providing an open-source code can compensate for the lack of description/justification, which is obviously a mistake according to the biostatistics community standards.

- The rationale motivating the algorithmic choices should be explicit and convincing. This is possibly where the article most fails (as a typical example, i.e. non exhaustively, the choice of the polynomial basis is motivated by the least scientific possible argument i.e. because it is the default one in the poly R package). Other lacks of rationale can be found in my former review, where many comments led to justifications in the rebuttal, which are longer than the corresponding changes in the main body of the article.

In addition, the authors insufficiently answered to three important comments:

- about the geometric mean/rank product: Contrarily to what the authors claimed, the rank product is not a standard way of combining different tests: it was published as a way to provide an approximate significance test for genes in replicated experiments: In this setting, the ranks of genes in various replicate samples are combined to yield an approximate significance value. This largely differs from the authors' use-case, where different tests are combined into a unique significance value. I understand the semantic slippage (from replicated observations to tests) and its intuition, but considering the light theoretical support of the rank product (it is essentially motivated by biological common sense and benefit from scarce statistical ground), transposing it to the combination of various tests (a task for which a large body of literature already exists) without further justification is in my opinion not sufficient wrt methodological publication standards.

- about the links between MV imputation and regression (comment 26): I have scrutinized the authors answer, which in my opinion essentially argues about semantic aspects, so that I stick to my former point of view.

- similarly, the authors' response to my comment 25, which despite involving many additional comparisons (among which some completely nonsensical, as those described in the last paragraph of the answer), failed to answer my question.

For all these reasons, and even though it possibly helped the authors to process their own datasets of interest, the article is such that:

(1) I cannot advise any of my wet-lab colleague (ie with very few knowledge in biostatistics) to use RoIDE methodology as a black-box, while trusting its result thanks to its solid methodological foundations;

(2) As a researcher in biostatistics, I cannot rely on the publication to discuss/improve/generalize the approach, as it is classically possible to do with the methodological publications I read.

Therefore, I stick to my former editorial advice, which is to reject this manuscript.

Reviewer: 2

Comments to the Authors

To complete this revised version, the authors performed an impressive amount of experimental work and tentatively answer most of my remarks from the first round. I acknowledge them for doing so, yet, the revised version failed to convince me in broadly the same terms as the first one. Although few of the authors' answers were highlighting, some raised new concerns and most confirmed my general opinion after reading the first version.

This now confirmed opinion is the following: the article describes (only partially and in vague literate terms in place of algorithms and formulae) a rather labyrinthine procedure, which various elements are weakly supported (either by theoretical results or by well-formulated rationale; as opposed to extensive empirical evaluations on simulated data). **It does not mean that it cannot deal with the task it was originally designed for, and in fact, it is probably the case. However, as smart as it is, it is not sufficient to be considered a methodological development in biostatistics.** This is the main difference between the engineering tricks and workflows that sometimes accompanies the M&M section of an article describing new biological discoveries, and a methodological development that is worth of publication as a self-contained novelty. Notably and as already explicated in my former review, in the latter case, the article must provide an exhaustive and accurate description/justification/evaluation of every single step.

The authors largely extended the volume of their simulations to address the last point, however: (1) it remains difficult to assess the extent to which these simulations fit with their algorithmic implicit assumptions (so that it is legitimate to wonder on the generalization capabilities on different real data), and (2) at some points repeated simulations cannot cope for the lack of theoretical guarantees. As for the two other points (description and justification), the article changes are not sufficient, notably:

Our response:

Overall, we feel that evaluating the proposed manuscript only from the perspective of methodological development in biostatistics is somewhat beside the point. Instead of introducing a new computational framework as a 'self-contained novelty', our aim was to present a pragmatic and realistic approach into detecting longitudinal differential expression in complex and often noisy proteomics data, where thousands of proteins are inspected simultaneously. We apologize if this has not been conveyed clearly enough in our manuscript. This is also far from an 'engineering trick', and potentially something with great practical impact. For experimentalist researcher, the interest is typically in ranking the proteins based on their strength of true differential expression. Thus, the ability of a given differential expression method to correctly detect true differential expression (and discard false findings) is of utmost importance to avoid unnecessary validation experiments.

Our study encompasses an extensive comparison of 16 approaches, including our novel RoIDE approach. The benchmarking includes several thousands of diverse types of semi-simulated datasets to test the generalization capabilities of the methods. These benchmarking datasets are only partly simulated; they are based on real experimental proteomics spike-in datasets, possess qualities of real experimental data and are therefore especially suitable for this kind of benchmarking. Furthermore, our benchmark also includes real experimental biological data, i.e., the new *Francisella tularensis* data with 180 samples, including both biological and technical replicates. The reproducibility and ability of the methods to produce biologically meaningful results are thoroughly scrutinized with this data. Importantly, the findings using this real experimental data were congruent with the semi-simulated benchmarking data.

Comment 1:

The list of parameters (those accessible to the users as well as those which can be tuned to a default value) still fails to appear in a synthetic way in the article; and the entire necessary information about their precise

influence, their individual stability, their range of acceptable values, etc. is only partially provided by extensive yet too global comparisons for which no clear material can be extracted.

Our response:

Besides the user manual, it is clearly mentioned in the discussion of the manuscript, that for the default usage of RolDE, the user does not need to provide **any other parameters** than the design matrix and whether the time points are aligned or not. The other adjustable parameters for the default function of RolDE are related to the number of CPU cores used, and to significance estimations (number of permutations, multiple hypothesis adjustment method), which **do not affect** the ordering of the proteins by the algorithm and, thus, do not affect the benchmarking outcomes. Additionally, for the expert user, we offer the possibility to customize various parameters, such as the polynomial degrees and the model type for the different modules, through an explicit function. The effect of these parameters to the performance of the algorithm was extensively covered in the revised manuscript (**Supplementary Figures 3**). We apologize for possible confusion if the available parameters were not stated clearly enough in the manuscript.

Comment 2:

The very details of the algorithm (possibly through the form of an algorithm or a detailed flowchart) is still missing. Worse, I discovered when reading the rebuttal that depending on the timepoint being align or not, or depending on the presence of MV, some subroutines in various modules may change their processing, in a way that is not described in the article. To me, this more a source of concern than anything else. Worst, in their answer, the authors seems to assume that providing an open-source code can compensate for the lack of description/justification, which is obviously a mistake according to the biostatistics community standards.

Our response:

To improve the understanding into the mechanisms of our RolDE approach, we have included a flowchart as **Supplementary Figure 9** in the revised manuscript. Indeed, the Reviewer is correct that in the case of non-aligned time points, RolDE adjusts its analysis approach. This is required to ensure comparability of the orthogonalized polynomials between the individuals in the RegROTS module and due to the unfeasibility of using aligned time points in the DiffROTS module. We **have** described these details in the manuscript (page 15, lines 34-40, page 17, lines 1-16). Suggesting that subroutines change their processing in a way that is not described in the article, **is incorrect**, as these variations to the modules are indeed well described. Furthermore, unlike the Reviewer claims, the algorithm **does not** change as a result of the data containing missing values or not.

Comment 3:

The rationale motivating the algorithmic choices should be explicit and convincing. This is possibly where the article most fails (as a typical example, i.e. non exhaustively, the choice of the polynomial basis is motivated by the least scientific possible argument i.e. because it is the default one in the poly R package). Other lacks of rationale can be found in my former review, where many comments led to justifications in the rebuttal, which are longer than the corresponding changes in the main body of the article.

Our response:

The original question of Reviewer 2 was: “What family of orthogonal polynomials are used in RegROTS?” We answered in our response letter: “The orthogonal polynomials in the RegROTS module are generated using the three term recursion algorithm described in [67, pp.343-4].” And in the manuscript: “The orthogonal polynomials are defined using the three term recursion algorithm described in [67, pp.343-4] and implemented in the function poly of R.” (67. Kennedy WJ, Gentle JE (1980) Statistical computing. Marcel Dekker Ltd)

As the exact way in which the orthogonal polynomials were defined was not important, using a very established, generally approved and used approach (introduced already four decades ago) should not be considered as “least scientific”. **Quite the contrary**; as this approach has been evaluated countless times, using this approach should be a very valid and strong choice if there are no specific reasons to apply an alternative approach. We are delighted to hear that the Reviewer was happy with the other justifications and we apologize if the corresponding changes in the manuscript were not long enough.

Comment 4:

About the geometric mean/rank product: Contrarily to what the authors claimed, the rank product is not a standard way of combining different tests: it was published as a way to provide an approximate significance test for genes in replicated experiments: In this setting, the ranks of genes in various replicate samples are combined to yield an approximate significance value. This largely differs from the authors' use-case, where different tests are combined into a unique significance value. I understand the semantic slippage (from replicated observations to tests) and its intuition, but considering the light theoretical support of the rank product (it is essentially motivated by biological common sense and benefit from scarce statistical ground), transposing it to the combination of various tests (a task for which a large body of literature already exists) without further justification is in my opinion not sufficient wrt methodological publication standards.

Our response:

While the Reviewer is correct in saying that “the rank product is not a standard way of combining different tests”, the rank product **has** been used in meta-analysis combining datasets from multiple studies and **has** been observed to perform well in comparison to more standard approaches such as Fisher's inverse chi-square method [Hong F, Breitling R (2008) *A comparison of meta-analysis methods for detecting differentially expressed genes in microarray experiments. Bioinformatics 24(3):374–382*, Campain A, Yang YH (2010) *Comparison study of microarray meta-analysis methods. BMC Bioinformatics 11(1):408. <https://doi.org/10.1186/1471-2105-11-408>*]. Furthermore, because RolDE operates with ranks, the rank product is a natural way of combining the different modules (tests); the significance values themselves are not directly combined but their ranks. As evident from our benchmarking in this manuscript and prior comparisons by others, the rank product method is robust and works very well in practice, especially in noisy complex data. In our opinion, a new approach should not be discarded simply for being new (instead of a classical approach) if it is shown to perform well and there are no theoretical obstacles for its usage, which for the rank product approach, to our knowledge, there are not.

Comment 5:

About the links between MV imputation and regression (comment 26): I have scrutinized the authors answer, which in my opinion essentially argues about semantic aspects, so that I stick to my former point of view.

Our response:

While we agree with the Reviewer that regression indeed can be useful in the presence of missing values, in RolDE it is **not** specifically used for imputation of values of any kind. In addition, as explained our previous response, the use of regression models in some form is also true for most of the methods evaluated in this study. Of the methods used in the *full* datasets with missing values, only Timecourse is not based on regression. Thus, most of the methods use regression modelling and should be able to deal with missing values in the data in a similar fashion to RolDE, resulting in a fair comparison even in the presence of missing values. Semantics aside, we would also like to re-emphasize that all the methods were also evaluated on altogether 810 datasets without any missing values.

Comment 6:

Similarly, the authors' response to my comment 25, which despite involving many additional comparisons (among which some completely nonsensical, as those described in the last paragraph of the answer), failed to answer my question.

Our response:

The original comment 25 by Reviewer 2:

“The lack of more refined comparisons and evaluation on the various subparts or the methodology makes it impossible to answer the following question: what is the important feature of the methodology? The intensive application of ROTS at any step of the method? Or ROTS is not that important, as the application of any other test on longitudinal models resulting from the proposed regressions would be equally good (which would indicate that the core of the proposed innovation is indeed the extensive use of regressed data)? Or, the combination of multiple ranks resulting from partial approaches (regardless of whether they are based on regression, or not; indicating that for longitudinal data, one should mainly perform tests in all possible directions, then combine them in a clever way)? Deeper comparisons should be conducted to answer these questions, as their answer would provide a necessary support to the presented methodological developments.”

In the revised manuscript, we demonstrated the importance and benefits of ROTS in the RegROTS module by substituting ROTS with ANOVA and showing that using ROTS results in considerable performance gains (Results section, page 6, lines 29-36, **Supplementary Figure 4**). Furthermore, in addition to comparing ROTS to ANOVA in the RegROTS module, we further demonstrated that the use of complementary approaches (multiple modules) in RolDE increased the performance further when compared to a single module (Results section, page 6, lines 37-41, **Supplementary Figure 4**).

Finally, through combining multiple existing approaches (Timecourse, Limma, MaSigPro) with a similar ranking and rank product approach as applied in RolDE, we demonstrated that combining multiple approaches in general can improve the performance when compared to the approaches separately, but not always (Results section, page 6, lines 41-47, page 7, lines 1-3, **Supplementary Figure 5**). Thus we have exactly answered to the original question of Reviewer 2 on the various subparts and demonstrated that: 1) ROTS is important and by simply substituting ROTS with any available test, would not lead to as good performance, 2) the combination of multiple approaches is important and can improve the performance further, but 3) the combination of any approaches does not automatically result in performance improvements, but should be considered carefully (as in RolDE with its diverse and well performing modules in detecting longitudinal differential expression).

Comment 7:

For all these reasons, and even though it possibly helped the authors to process their own datasets of interest, the article is such that:

- (1) I cannot advise any of my wet-lab colleague (ie with very few knowledge in biostatistics) to use RolDE methodology as a black-box, while trusting its result thanks to its solid methodological foundations;
 - (2) As a researcher in biostatistics, I cannot rely on the publication to discuss/improve/generalize the approach, as it is classically possible to do with the methodological publications I read.
- Therefore, I stick to my former editorial advice, which is to reject this manuscript.

Our response:

We would not recommend any of our colleagues (wet-lab or other) to use any statistical or computational tool without basic understanding about the methodology. In general, successful analysis of complex omics data requires specialized expertise.

We would also like to highlight that the proposed method is not intended to be a black-box method generalizable to any type of data. Instead, it is specifically intended as a discovery tool for longitudinal differential expression patterns from complex and often noisy omics data, especially proteomics data. The focus is on practical utility of the method in ranking the best candidate proteins at the top and discarding false findings. This is of utmost importance to avoid unnecessary validation experiments in a typical real-life

biological or biomedical study, where only the top candidates are usually validated via other more targeted approaches, such as targeted MS or western blotting. Due to the nature of typical proteomics data, the conventional statistical methods are often suboptimal in this setting, as demonstrated by our benchmarking. We have now further emphasized the need for validation of the top findings in the revised manuscript (page 11, lines 31-33).

REVIEWER COMMENTS

Reviewer #3 (Remarks to the Author):

In short, a revision of this paper is more likely to satisfy a benchmarking paper than a novel methodology. A couple of small suggestive changes to the MS include

1) Refocusing the introduction away from novel methods of innovation. Line 86 discussed the gap in the lack of comparison studies, and line 93 mentioned a new method. This slight misalignment is confusing.

2) Other key issue is the definition of differential expression in longitudinal data. Often, this is defined as the existence of one or more differences at any time point and the actual pattern is annotated later. The claim that “user does not need to provide RoDE any prior information on the type of differential expression patterns (e.g. stable, linear, quadratic, cubic, sigmoid, etc.)” doesn’t sit well with me. It is better to articulate the NULL hypothesis by defining each intended “Longitudinal DE” or simply identify ANY “longitudinal DE”. The evaluation method could compare how the various methods are able to pick up different types of DE patterns.

Reviewer: 3

Comments from the Editor and Reviewer:

You will see from Reviewer#3's comments pasted below, that this reviewer finds the paper suitable for publication in Nature Communications, provided that it can be modified to focus on the **benchmarking** aspects rather than the method development aspects.

In addition to the minor comments pasted below, Reviewer#3 mentioned to us that - for a benchmarking-focused paper to be convincing - it will be important to **include additional experimental data** (original or previously published) since the tool comparison is currently too much focused on simulated and semi-simulated data. Furthermore, Rev#2 says that additional **evidence for the practical utility of your approach** is required (i.e. that the implementation-related benefits of ROIDE mentioned in response to Rev#2 have a positive practical impact when applied to experimental data).

Our response:

We thank the Reviewer for these guiding suggestions and have now completely modified the manuscript, changing the focus from method development aspects into a comprehensive comparison and benchmarking of altogether 15 longitudinal methods and a baseline two group method in detecting differential expression in noisy longitudinal proteomics data, with and without missing values.

Furthermore, as suggested by the Reviewer, we have now included an additional experimental proteomics dataset and further extended our comparisons using this dataset. The new dataset consists of a previously published proteomics data³¹ from *in vitro* induced T regulatory cells (iTreg) using two different protocols for the T regulatory cell (Treg) induction together with mock-stimulated naïve T cells serving as controls. This new experimental dataset includes almost 10000 proteins, four time points, and two comparisons (Treg cells induced using the two different protocols vs. the mock-stimulated naïve T cells), in which the methods were compared.

In this new iTreg dataset, the ability of the methods to produce biologically meaningful results was extensively evaluated. As the differentiation and *in vitro* induction of naïve stimulated T cells into T regulatory cells is a complex process and the proteins afflicted by this process are still under research³¹⁻³³ and not known in detail, several Treg related gene sets were used to assess the biological relevance of the findings, providing a good surrogate of what should be expected to be affected during the induction process. For a comprehensive evaluation, altogether 28 Treg related human gene sets were downloaded from the Molecular Signatures Database (MSigDB) and combined into 14 Treg gene sets. In addition, a Treg signature gene set³³ from a large human study, genes from the Reactome pathway "*RUNX1 and FOXP3 control the development of regulatory T lymphocytes (Tregs)*", and the interactome of the forkhead box P3 (FOXP3) protein, which is a lineage specification factor of Treg cells, were used to explore Treg related functional enrichment. The new results in the experimental iTreg dataset are shown in **Figure 4A**. All the used gene sets are given in the new **Supplementary Table 2**. These results demonstrated how RolDE, BaselineROTS, Timecourse and Limma were able to provide biologically meaningful results also in another biological data set in addition to the *Fn* data.

In addition to exploring the biological relevance of the findings, the new iTreg data was used to explore the overlaps of the findings between the methods. Common findings by multiple methods from the same dataset are often considered more reliable than unique findings by a single method^{11,31}. For instance, in a prior

comparative study of differential expression methods in RNA-seq, it has been observed that true positive detections are generally shared by the methods, while the unique findings by a single method are rarely true positive detections¹¹. The overlap of the top 1000 findings of the methods are shown in new **Figures 4B-C**. The results confirmed the observations from the other comparisons that the top findings of RolDE can be considered reliable and are likely to be true positive detections, as they are typically also shared by many other methods.

The additional experimental iTreg dataset now further demonstrates the practical utility of RolDE. It was among the methods with highest enrichment of expected Treg related genes, supporting the biological relevance of the findings (**Figure 4A**). In addition, it was among the methods with largest proportions of shared top detections with the other methods, supporting the reliability of the findings (**Figure 4B-C**). In summary, RolDE performed best in the semi-simulated spike-in datasets (**Figure 2**), had good reproducibility between the technical replicated in the experimental *Fn* dataset (**Figure 3A-B**), and was among the top methods in ranking the results in a biologically meaningful way in the experimental *Fn* and iTreg datasets (*Fn*: **Figure 3C-D**, iTreg: **Figure 4A**). Furthermore, RolDE displayed most balanced performance of all the methods in detecting diverse trend differences of all possible types with good consistency (**Supplementary Figure 1**). The ability of RolDE to produce meaningful findings even with non-aligned timepoints was further demonstrated using a previously published longitudinal type 1 diabetes proteomics dataset with human individuals (**Figure 5**). In addition to good all-round performance, the default application and use of RolDE is very easy, which is now further demonstrated for the iTreg dataset in the new **Supplementary File 1 (Supplementary_File_Running_RolDE iTreg_Data.R)**. Taken together, these results clearly demonstrate the robustness and practical utility and applicability of RolDE in detecting, in principle, any kind of longitudinal differential expression in variable datasets and experimental settings.

Specific comments from the Reviewer

In short, a revision of this paper is more likely to satisfy a benchmarking paper than a novel methodology. A couple of small suggestive changes to the MS include

Comment 1: Refocusing the introduction away from novel methods of innovation. Line 86 discussed the gap in the lack of comparison studies, and line 93 mentioned a new method. This slight misalignment is confusing.

Our response: We thank the Reviewer for pointing this out. We have now revised the whole manuscript to fit the modified scope of a comprehensive comparison and benchmarking of longitudinal differential expression methods for proteomics data as suggested by the Reviewer (**page 3, lines 5-21**).

Comment 2: Other key issue is the definition of differential expression in longitudinal data. Often, this is defined as the existence of one or more differences at any time point and the actual pattern is annotated later. The claim that “user does not need to provide RolDE any prior information on the type of differential expression patterns (e.g. stable, linear, quadratic, cubic, sigmoid, etc.)” doesn’t sit well with me. It is better to articulate the NULL hypothesis by defining each intended “Longitudinal DE” or simply identify ANY “longitudinal DE”. The evaluation method could compare how the various methods are able to pick up different types of DE patterns.

Our response: We thank the Reviewer for this insightful observation. We agree that perhaps our choice of wording was unclear. Indeed, sometimes with certain methods the researcher should define the searched longitudinal differential expression pattern *a priori*, as Liu et al.¹⁰ In certain scenarios, this can also be

beneficial, if the interest is particularly in specific kinds of longitudinal expression patterns and pattern differences. However, as the Reviewer suggests, the focus of this manuscript was to detect any kind of differential expression between the conditions (including linear and non-linear trend differences in longitudinal protein expression, stable differences in expression over time, etc.) without prior knowledge of the type of differences investigated from the data. In **Supplementary Figure 1**, we have demonstrated how the different methods pick up longitudinal differential expression of specific kinds in different datasets. With RolDE, any kind of longitudinal differential expression can be investigated without the user needing to specify *a priori* what kind of differential expression patterns is to be searched from the data or what polynomial degrees should be used, unlike with many of the regression-based approaches. Thus, by default RolDE effectively identifies any type of longitudinal differential expression (**Supplementary Figure 1**). We have now further clarified this in the revised manuscript (**page 11, lines 47-48, page 12, line 1**).

REVIEWERS' COMMENTS

Reviewer #3 (Remarks to the Author):

I thank the authors for their responses, and the newly revised manuscript is much improved. They have successfully refocused the paper to a benchmarking style study. The authors have addressed point 1 raised in the previous review but I didn't feel they have addressed my point in point 2 satisfactorily.

The author continues to claim that the "user does not need to provide RoIDE any prior information ..." as a key innovation and stresses the limitations of the regression-based model. I think is better to phrase that your model handles an unknown function. On this point, the framework should be best compared to the concept behind the "generalized additive model" (GAM) and suggests the authors refer to the extensive literature around GAM where the response is model against "unknown smooth functions of some predictor variables". I feel strongly that such rephrasing throughout the MS is essential. In particular, the last sentence of the "abstract".

Reviewer: 3

Comment:

I thank the authors for their responses, and the newly revised manuscript is much improved. They have successfully refocused the paper to a benchmarking style study. The authors have addressed point 1 raised in the previous review but I didn't feel they have addressed my point in point 2 satisfactorily.

The author continues to claim that the "user does not need to provide RolDE any prior information ..." as a key innovation and stresses the limitations of the regression-based model. I think is better to phrase that your model handles an unknown function. On this point, the framework should be best compared to the concept behind the "generalized additive model" (GAM) and suggests the authors refer to the extensive literature around GAM where the response is model against "unknown smooth functions of some predictor variables". I feel strongly that such rephrasing throughout the MS is essential. In particular, the last sentence of the "abstract".

Our response:

We thank the Reviewer for the suggestion. We have now modified the last sentence of the abstract as suggested by the Reviewer on **page 1 lines 29-31**:

Furthermore, RolDE is suitable for different types of data with typically unknown patterns in longitudinal expression and can easily be applied even by non-experienced users.

Furthermore, as the Reviewer suggests, we have added discussion comparing RolDE to the GAM framework on **page 12, lines 25-32**, and related **new references 60 and 61**:

Similar to the generalized additive modeling (GAM) framework^{60,61}, where the response variable is modeled against unknown smooth functions of explanatory variables, RolDE can simultaneously search for many different types of patterns related to longitudinal differential expression. While the advantage of GAM is that it is not limited to global parametric functions, such as polynomials, which provides flexibility to adapt the fit to the data even with complex non-linear relationships, its downside is the propensity to overfit. Therefore, with RolDE we decided to focus on relatively simple models, which is crucial for typical longitudinal proteomics studies, involving small to moderate numbers of individuals and time points.

60. Rigby, R. A. & Stasinopoulos, D. M. Generalized additive models for location, scale and shape. J R Stat Soc Ser C Appl Stat 54, 507–554 (2005).

61. Hastie, T. & Tibshirani, R. Generalized additive models. Stat Sci 1, 297–310 (1986).